# SHINE: A Scalable In-Context Hypernetwork for Mapping Context to LoRA in a Single Pass

**Yewei Liu** [1 2]  **Xiyuan Wang** [1]  **Yansheng Mao** [1 2]  **Yoav Gelberg** [3]  **Haggai Maron** [4]  **Muhan Zhang** [1]

## Abstract

We propose SHINE (Scalable Hyper In-context NEtwork), a scalable hypernetwork that can map diverse meaningful contexts into high-quality LoRA adapters for large language models (LLMs). By reusing the frozen LLM's own parameters in an in-context hypernetwork design and introducing architectural innovations, SHINE overcomes key limitations of prior hypernetworks and achieves strong expressive power with a relatively small number of parameters. We introduce a pretraining and instruction fine-tuning pipeline, and train our hypernetwork to generate high quality LoRA adapters from diverse meaningful contexts in a single forward pass. It updates LLM parameters without any fine-tuning, and immediately enables complex question answering tasks related to the context without directly accessing the context, effectively transforming in-context knowledge to in-parameter knowledge in one pass. Our work achieves outstanding results on various tasks, greatly saves time, computation and memory costs compared to SFT-based LLM adaptation, and shows great potential for scaling. Our code is available at https://github.com/MuLabPKU/SHINE

## 1. Introduction

Large language models (LLMs) have established themselves as a cornerstone of modern artificial intelligence (Vaswani et al., 2017). Adapting these pretrained models to downstream tasks is typically approached through either prompt engineering (Brown et al., 2020) or parameter fine-tuning (Ouyang et al., 2022). However, both paradigms face

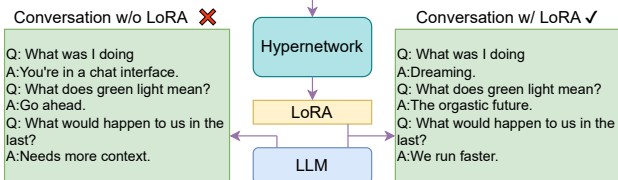

Context

And as I sat there brooding on the old, unknown world, I thought of Gatsby's wonder when he first picked out the green light on the end of Daisy's dock. He had come a long way to this blue lawn, and his dream must have seemed so close that he could hardly fail to grasp it. He did not know that it was already behind him, somewhere back in that vast obscurity beyond the city, where the dark fields of the republic rolled on under the night. Gatsby believed in the green light, the orgastic future that year by year recedes before us. It eluded us then, but that's no matter, tomorrow we will run faster, stretch out our arms farther… and one fine morning-So we beat on, boats against the current, borne back ceaselessly into the past.

Conversation w/o LoRA ✗

Q: What was I doing
A:You're in a chat interface.
Q: What does green light mean?
A:Go ahead.
Q: What would happen to us in the last?
A:Needs more context.

Hypernetwork

LoRA

LLM

Conversation w/ LoRA ✓

Q: What was I doing
A:Dreaming.
Q: What does green light mean?
A:The orgastic future.
Q: What would happen to us in the last?
A:We run faster.

*Figure 1.* **An Example of SHINE:** It maps context to LoRA in a single pass without any fine-tuning. The LoRA can be used for downstream conversation without accessing the context.

significant constraints. Prompt-based methods exacerbate inference latency and consume valuable context window capacity, while fine-tuning entails substantial training overhead, sensitivity to hyperparameters, and the storage burden of maintaining distinct model parameters for each task.

The advances in hypernetworks offer a promising third alternative (Ha et al., 2017; Chauhan et al., 2024). A hypernetwork is a neural network designed to generate the weights for another network. By leveraging a hypernetwork to generate adapters for an LLM (Mahabadi et al., 2021), adaptation can be achieved in a **single forward pass**, eliminating the need for additional prompts or additional training.

Despite this appealing vision, applying hypernetworks to LLMs presents significant challenges. The parameter space of modern LLMs is extremely large, making parameter generation for all modules in LLM prohibitively expensive. As a result, prior approaches make many compromises like only generating LoRAs for a subset of layers (Chen et al., 2025), using a very small bottleneck in the hypernetwork design (Jukic et al., 2025), or reusing a small MLP multiple times (Charakorn et al., 2025; Xiao et al., 2023; Abdalla et al., 2025). These constraints have so far prevented hypernetworks from becoming a practical and scalable solution for LLM adaptation.

In this work, we address these challenges by proposing a novel hypernetwork architecture and a full pretraining and instruction fine-tuning pipeline to "meta-train" the hypernet-

[1]Institute for Artificial Intelligence, Peking University [2]School of Electronics Engineering and Computer Science, Peking University [3]University of Oxford [4]Technion, NVIDIA. Correspondence to: Muhan Zhang <muhan@pku.edu.cn>.

*Proceedings of the 43rd International Conference on Machine Learning*, Seoul, South Korea. PMLR 306, 2026. Copyright 2026 by the author(s).

work. Using an in-context design, we adapt the LLM itself as part of the hypernetwork. Instead of MLP, a lightweight Transformer is used to exchange messages between layers and to generate the corresponding LoRAs for each layer. Our architecture design has no bottleneck, maintains strong expressive capacity and only trains a relatively small number of parameters. After training, our hypernetwork SHINE (Scalable Hyper In-context NEtwork) can generate high quality LoRAs (Hu et al., 2022) directly from a given meaningful context in one forward pass, without any gradient-based optimization. The generated LoRAs can be used for downstream conversation and question answering without accessing the context, effectively transferring the contextual knowledge into model parameters (Figure 1).

Our contributions are summarized as follows:

- We propose a new bottleneck-free hypernetwork architecture, SHINE, for LLM adaptation that achieves high expressive power using a small number of parameters.
- We introduce a pretraining and instruction fine-tuning pipeline that enables the hypernetwork to generate high quality LoRA adapters from diverse contexts in a single forward pass, without any test-time optimization.
- We utilized a dataset comprising 6 billion pre-training tokens, combined with carefully collected instruction tuning data—at a scale far exceeding all previous hypernetwork implementations. Leveraging the expressivity and scaling potential inherent in our architecture, our hypernetwork achieves superior results significantly outperforming baselines. Additionally, it shows no sign of hitting capacity bottleneck with consistent performance gains in the scaling experiments, highlighting its potential for large-scale training and practical deployment.

## 2. Related Works

**Hypernetwork & Metanetwork.** Hypernetwork (Ha et al., 2017) or metanetwork (Munkhdalai & Yu, 2017) are networks over networks. They themselves are networks, but can take other network parameters (Zhou et al., 2023; Putterman et al., 2024), gradients (Gelberg et al., 2025), features and relative contexts as input and output network features (Unterthiner et al., 2020), new weights (Navon et al., 2023; Zhou et al., 2023; Lim et al., 2024; Kofinas et al., 2024) or gradients (Finn et al., 2017; Andrychowicz et al., 2016), etc.

Hypernetworks and metanetworks have emerged as powerful paradigms for analyses and generation on neural networks. By automating the process in a meta-learning way (Hospedales et al., 2022), they eliminate the need for manual design, consistently achieving better results compared to heuristic-based approaches. They have achieved significant progress in fields like network pruning (Liu et al.,

2019; 2025), reinforcement learning (Beck et al., 2022; Sarafian et al., 2021), neural architecture search (Zhang et al., 2019; Peng et al., 2022), model editing (Mitchell et al., 2022; Tan et al., 2024), and LLM adaptation.

**Hypernetwork for LLM Adaptation.** Hypernetwork-based LLM adaptation represents a promising new frontier. By bypassing both the iterative overhead of fine-tuning and the context-window constraints of in-context learning, hypernetworks generate task-specific adapters in a single forward pass, offering a significant leap in efficiency. Due to the high dimensionality of LLMs, recent research has converged on the generation of LoRA (Hu et al., 2022; Back et al., 2026) adapters. Following this trajectory, our work focuses exclusively on the efficient generation of LoRA adapters. We note that a small number of hypernetwork-based approaches generate different adaptation parameters. For example, Ji et al. (2026) predicts gating coefficients $\beta$ within SwiGLU blocks rather than LoRA weights.

Designing effective hypernetworks also remains a formidable challenge due to the extreme dimensionality of the target parameter space. Existing approaches (Charakorn et al., 2025; Xiao et al., 2023; Abdalla et al., 2025; Liao et al., 2025; Walsh et al., 2026; Park et al., 2026) typically mitigate this by employing a small MLP to generate weight segments independently, which are then concatenated to form the final adapter. We argue that this "segment-wise" generation treats weights in isolation, failing to capture the global dependencies and latent structural correlations essential for coherent weight adaptation. Consequently, such methods may yield suboptimal adapters that lack the global coordination required for complex tasks.

To bridge this gap, we propose a fully Transformer-based hypernetwork that leverages self-attention to facilitate superior information exchange and global context modeling. Adopting an in-context framework akin to (Ge et al., 2024; Chen et al., 2025; Jukic et al., 2025; Charakorn et al., 2026; Phang, 2024), we utilize the pre-trained LLM's own internal representations to predict its LoRA parameters. This strategy exploits the rich, pre-trained inductive bias of the LLM (Delétang et al., 2024), enhancing expressivity without necessitating extensive training from scratch. Crucially, unlike prior in-context approaches, our architecture introduces a novel mechanism that enables holistic, bidirectional communication across the parameter space, achieving high expressivity with a relatively small number of parameters.

We provide a detailed analysis and comparison with some prior works in Appendix C.

**Test-Time Training (TTT).** Test-Time Training (TTT) is closely related to our approach, as both methods update model parameters at inference time. The key difference lies

in how the update is obtained: our method uses a hypernetwork to directly predict parameter changes in a single forward pass, whereas TTT typically derives such changes through gradient-based optimization at test time. While our approach is still an early exploration, TTT has been extensively studied for LLMs (Sun et al., 2025; Tandon et al., 2025; Feng et al., 2026) and has also been extended to vision tasks (Han et al., 2025; Li et al., 2026). A recurring insight in TTT-based work is that the KV cache can be compressed or adapted. For example, Gelberg et al. (2026) explicitly compress the KV cache through continual training with masks.

**Hypernetworks for Different Architectures.** Beyond MLPs and Transformers, hypernetwork-style parameter generation has been explored for a wide range of architectures. In generative models, diffusion models have been used to generate LoRA adapters (Xiao et al., 2025) or even full model parameters (Wang et al., 2025). Other than full attention, Han (2026) generate LoRA adapters for MoE architectures (Shazeer et al., 2017). Related ideas have also been used to dynamically generate weights that replace Transformer attention (He et al., 2026). In vision and multimodal settings, hypernetworks have been applied to video semantic control (Wu et al., 2026), video understanding (Nguyen et al., 2026), text-guided image editing (Team, 2026), and vision-language models (Shao et al., 2025b). More generally, Zhou (2026) propose a hypernetwork framework for arbitrary architectures.

**Hypernetworks for Additional Tasks and Analysis.** Hypernetworks have also been studied for diverse adaptation and analysis tasks. Tan et al. (2025a) combine hypernetworks with parametric RAG to selectively retrieve knowledge and transform it into model parameters. Tan et al. (2025b) apply hypernetworks to cross-scale knowledge transfer, while Shao et al. (2025a) use them to merge LoRA adapters. Similar mechanisms for changing parameters have been shown to be important and have been explored in agentic settings. (Bao et al., 2026; Xu et al., 2026; Kumar, 2026). Instead of directly generating a full LoRA adapter, Vaidya et al. (2026) use a hypernetwork to route over shared LoRA ranks, thereby controlling model behavior. Finally, Cheng et al. (2026) analyze conflicts between hypernetwork-generated LoRA adapters and the base LLM, providing insight into failure modes of hypernetwork-based adaptation.

## 3. Architecture

### 3.1. Challenges in Prior Hypernetwork Design

Our objective is to build a hypernetwork capable of directly generating LoRA adapters for LLMs. This task presents three fundamental challenges:

- **Semantic-to-Parameter Alignment:** The hypernetwork must bridge the significant gap between natural language input and weight space. It requires strong language understanding to translate complex text input into precise functional adjustments in the adapter weights.
- **High-Dimensional Output:** Even within a low-rank setting, mapping to the full set of LoRA weights for all LLM layers imposes a significant burden on the architectural design, requiring strong expressivity of the hypernetwork.
- **Efficiency:** To be a viable alternative to standard fine-tuning, the hypernetwork must generate adapters with minimal latency and computational overhead, facilitating rapid switching between tasks.

Prior approaches typically fail to address these challenges simultaneously. They either employ suboptimal architectures that do not scale well and can only generate a subset of LoRAs—or rely on restrictive bottlenecks (e.g., reusing small MLPs), which severely limits expressivity and confines the model to simple tasks.

### 3.2. Overall Architecture

We propose a novel architecture that effectively addresses both expressivity and efficiency. To ensure robust language understanding without introducing external encoders, we leverage the LLM backbone itself to compress input context into memory tokens. To capture the full spectrum of semantic information—ranging from low-level syntax to high-level reasoning—we collect representations from all layers of the model. Furthermore, we address the challenge of high-dimensional output by eschewing standard, parameter-inefficient wide MLPs. Instead, we treat the memory states as a token sequence and train a lightweight Transformer to facilitate message passing among the memory states.

As illustrated in Figure 2, our method proceeds as follows:

1. **Memory Extraction:** Previous work generates LoRA from **a single state** of the entire context (Charakorn et al., 2025), which introduces severe bottlenecks. We introduce *Meta LoRA* and append a sequence of learnable *memory embeddings* to the input text. The LLM with Meta LoRA processes this sequence in a single forward pass, compressing the context into hidden states of **a sequence of memory tokens** (we call "memory states"), which greatly widens the bottleneck between context and parameters.

2. **LoRA Generation:** With the memory states as input, a specialized M2P (memory to parameter) Transformer performs self-attention among multi-layer states and predicts the target LoRA parameters. Notably, the self-attention is decomposed into alternate row and column **bidirectional** attentions to improve efficiency and enable deep layers to pass information back to shallow layers.

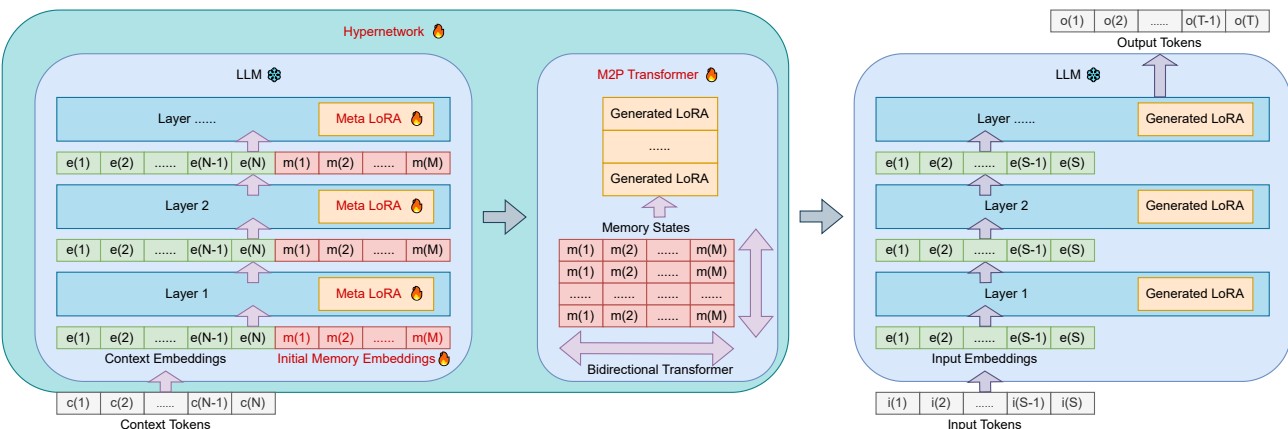

*Figure 2.* **Overall Architecture.** The process consists of two passes: (1) *Memory Extraction*, where the LLM (augmented with Meta LoRA) processes context to produce memory states, and (2) *Parameter Generation*, where a hypernetwork converts these states into task-specific LoRA adapters for the final inference.

3. **Downstream Task:** Finally, the LLM with the generated LoRA are used for the task based on the context.

The data flow is summarized as follows:

$$\text{Context} \xrightarrow{\text{LLM + Meta LoRA}} \text{Memory States}$$

$$\text{Memory States} \xrightarrow{\text{M2P Transformer}} \text{Generated LoRA}$$

$$\text{Question} \xrightarrow{\text{LLM + Generated LoRA}} \text{Answer}$$

### 3.3. Memory Extraction

Consider a backbone Transformer with $L$ layers and hidden dimension $H$. Let the input consist of context token embeddings $\mathbf{X}$ of length $N$ and learnable memory embeddings $\mathbf{M}_0$ of length $M$. The input sequence is defined as:

$$\mathbf{h}_0 = [\mathbf{X}; \mathbf{M}_0] \in \mathbb{R}^{(N+M) \times H} \tag{1}$$

We then feed the input sequence to the backbone LLM equipped with Meta LoRA. Let $\mathbf{h}_i$ denote the output hidden states of the $i$-th layer. We extract the memory states corresponding to the last $M$ tokens from each layer:

$$\mathbf{M}_i = \mathbf{h}_i[N:N+M,:] \in \mathbb{R}^{M \times H}, \quad i = 1, \ldots, L. \tag{2}$$

We stack all $\mathbf{M}_i$ into a global memory tensor:

$$\mathbf{M} = \text{Stack}([\mathbf{M}_1, \ldots, \mathbf{M}_L]) \in \mathbb{R}^{L \times M \times H}. \tag{3}$$

To ensure that the memory tensor contains enough information for the generated LoRA, we choose hyperparameter $M$ according to the rank of the generated LoRA. Let $r$ be the generated LoRA rank and $D$ be the sum of input and output dimensions of all linear layers in a single LLM layer. The number of LoRA parameters is $rD$. We enforce the memory length $M$ such that:

$$M = \lceil \frac{rD}{H} \rceil. \tag{4}$$

Then the size of memory states is larger than or equal to that of LoRA parameters.

### 3.4. M2P Transformer

The mapping from memory states to LoRA is designed as a lightweight Transformer with three steps (See Figure 3).

**Step 1: Positional Encoding.** To make the Transformer aware of memory states' layer and token sequence structure, we propose learnable positional embeddings based on the layer index and the memory token index:

$$\mathbf{P}^{\text{layer}} \in \mathbb{R}^{L \times 1 \times H}, \quad \mathbf{P}^{\text{token}} \in \mathbb{R}^{1 \times M \times H}, \tag{5}$$

$$\tilde{\mathbf{M}} = \mathbf{M} + \mathbf{P}^{\text{layer}} + \mathbf{P}^{\text{token}}, \tag{6}$$

where broadcasting is applied to match dimensions.

**Step 2: Sparse Attention Transformer.** We employ a lightweight Transformer to process $\tilde{\mathbf{M}}$. Flattening $\tilde{\mathbf{M}}$ into a sequence of length $L \cdot M$ would incur a prohibitive $O((LM)^2)$ self-attention cost. Instead, we adopt a sparse attention strategy that alternates between mixing information across layers (column attention) and across memory tokens (row attention). This strategy leads to $O(LM^2 + ML^2)$ time complexity and can save up to 90% flops compared with full attention in our experiment, as shown in Appendix B.5.

Let $\mathbf{Z}^{(i)} \in \mathbb{R}^{L \times M \times H}$ and $\mathbf{Z}^{(i+1)} \in \mathbb{R}^{L \times M \times H}$ denote the input and output of the $i$-th M2P Transformer layer, respectively, with $\mathbf{Z}^{(1)} = \tilde{\mathbf{M}}$. Let $\mathbf{Y}^{(i)} \in \mathbb{R}^{L \times M \times H}$ denote attention output at the $i$-th layer.

We apply column attention and row attention alternately, with odd layers using column attention and even layers using row attention. If $i$ is odd, the column attention proceeds as:

$$\mathbf{Y}^{(i)}_{:,j} = \text{SelfAttn}(\mathbf{Z}^{(i)}_{:,j}), \text{ for } j = 1, 2, ..., M, \tag{7}$$

where SelfAttn is the standard bidirectional self-attention layer. If $i$ is even, the row attention proceeds as:

$$\mathbf{Y}^{(i)}_j = \text{SelfAttn}(\mathbf{Z}^{(i)}_j), \text{ for } j = 1, 2, ..., L. \tag{8}$$

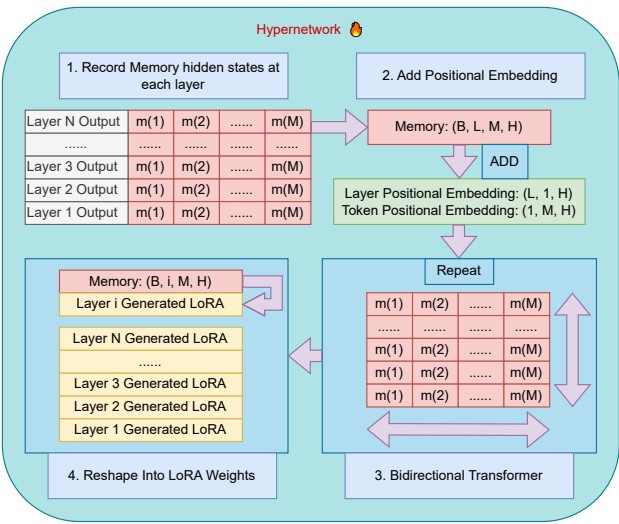

*Figure 3.* **Hypernetwork Architecture.** The model uses alternating attention along layer and token axes to efficiently process the memory tensor before projecting it into weights.

After self-attention, all layers use the same 2-layer MLP to update each token embedding:

$$\mathbf{Z}_{jk}^{(i+1)} = \mathrm{MLP}(\mathbf{Y}_{jk}^{(i)}), \text{ for } j = 1, 2, ..., L, k = 1, 2, ..., M. \quad (9)$$

**Step 3: Parameter Generation.** Let the final layer output from the M2P Transformer be $\hat{\mathbf{M}} \in \mathbb{R}^{L \times M \times H}$ (same shape as $\mathbf{M}$). Our goal is to transform it into LoRA weights. We partition the tensor such that the $i$-th slice $\hat{\mathbf{M}}[i, :, :]$ generates parameters for the $i$-th layer of the LLM.

To generate the weights, we flatten the memory slice for the specific layer into a vector $\mathbf{v} \in \mathbb{R}^{M \cdot H}$. We then sequentially slice and reshape $\mathbf{v}$ into different LoRAs. Suppose the first $t$ elements in $\mathbf{v}$ have been used. To generate LoRA for $\mathbf{W} \in \mathbb{R}^{I \times O}$, where $I$ is the input feature dimension and $O$ is the output feature dimension, we calculate:

$$\mathbf{A} = \mathrm{Reshape}(\mathbf{v}[t : t + I \cdot r]) \in \mathbb{R}^{I \times r}$$

$$\mathbf{B} = \mathrm{Reshape}(\mathbf{v}[t + I \cdot r : t + I \cdot r + r \cdot O]) \in \mathbb{R}^{r \times O}$$

Then $t$ is updated with $t + Ir + rO$ and used to generate the next LoRA. We discuss some alternative parameter generation approaches in Appendix A.2.

### 3.5. Advantages of Our Architecture

Our design solves all three challenges:

- **Semantic-to-Parameter Alignment:** Instead of training an external encoder from scratch, we leverage the pretrained LLM backbone itself for context encoding. This allows our hypernetwork to inherit the backbone's robust language comprehension capabilities, ensuring that the input context is mapped to a feature space that is already aligned with the model's internal representations.
- **High-Dimensional Output:** Our architecture ensures expressivity and parameter efficiency through an M2P

Transformer with bidirectional row/column attention. While prior works (Chen et al., 2025; Charakorn et al., 2026) also generate LoRA weights using hidden states, their hidden states in the $i$-th layer only contain information from lower ($j \leq i$) layers. Our approach mixes memory states **bidirectionally**, across both layers and tokens. This allows information to flow **from deep layers back to shallow layers**. Intuitively, this design mimics the **backpropagation** process, where *parameter updates in shallow layers are mathematically dependent on those of deeper layers*. By enabling this global information flow, we avoid the "blindness" of layer-wise generation.

- **Efficiency:** Our architecture avoids using wide linear layers or MLPs as in previous work (Chen et al., 2025), but uses a lightweight Transformer over memory tokens, which preserves computational efficiency while maintaining strong expressive capacity. Empirically, SHINE introduces negligible adaptation overhead compared to SFT. It also reduces inference cost compared with ICL because the contextual information is stored in LoRA, eliminating the need to include it in the input.

A further analysis of SHINE's expressive power compared to alternative architectures is in Appendix C.

## 4. Training

Our primary objective is to **learn a hypernetwork capable of mapping any meaningful natural language contexts to high-quality LoRA adapters**. We define meaningful contexts as coherent, semantically well-formed text. A high-quality adapter is defined as one that enables the base model to answer context-dependent queries naturally and accurately without directly accessing the context.

While conceptually straightforward, high quality context QA datasets are rare. Therefore, our training pipeline mirrors the standard LLM development lifecycle, consisting of two stages: **Pretraining** on general language modeling tasks and **Instruction Fine-Tuning** on QA tasks.

**Notation:** Let $\mathbf{c} = (c_1, c_2, \ldots, c_N)$ denote the input context. The trainable parameters include the M2P Transformer ($\Theta_{\mathrm{T}}$), the Meta-LoRA parameters ($\Theta_{\mathrm{M}}$), and the initial memory embeddings ($\mathbf{m}$). The frozen base LLM parameters are denoted as $\Theta_{\mathrm{LLM}}$. The generated LoRA adapter, denoted as $\Theta_{\mathrm{GLoRA}}$, is a function of the context and the hypernetwork: $\Theta_{\mathrm{GLoRA}} = f(\mathbf{c}; \Theta_{\mathrm{T}}, \Theta_{\mathrm{M}}, \mathbf{m})$. Let $P(\mathbf{y}|\mathbf{x}; \Theta_{\mathrm{GLoRA}}, \Theta_{\mathrm{LLM}})$ denote the probability that LLM with LoRA $\Theta_{\mathrm{GLoRA}}$ generates text $\mathbf{y}$ with input prompt $\mathbf{x}$.

### 4.1. Pretraining

Pretraining establishes the hypernetwork's ability to compress and reconstruct context. We train the model in a self-supervised manner on a large corpus using two comple-

mentary objectives: **Reconstruction** and **Completion**.

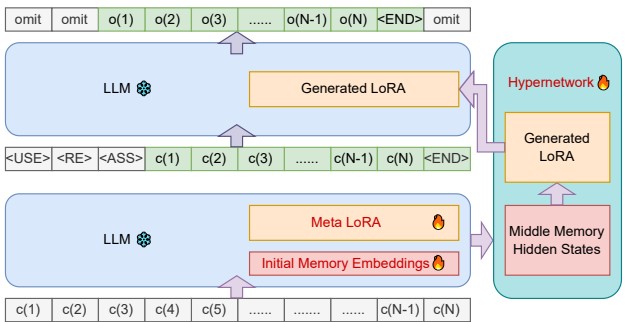

*Figure 4.* **Reconstruction Task:** The hypernetwork encodes the full context into a LoRA. The LLM is then prompted to reconstruct the original text.

**Reconstruction.** The reconstruction task ensures that the generated LoRA retains complete information about the input context (Figure 4). In each iteration, the hypernetwork converts a context $\mathbf{c}$ into $\Theta_{\mathrm{GLoRA}}$. The LLM, equipped with $\Theta_{\mathrm{GLoRA}}$, is prompted with text command `<RECON>` and trained to output the original context $\mathbf{c}$. The training loss is cross entropy loss:

$$\mathcal{J}_{\mathrm{RECON}} = -\log P(\mathbf{c}\,|\,\texttt{<RECON>}; \Theta_{\mathrm{GLoRA}}, \Theta_{\mathrm{LLM}}). \quad (10)$$

**Completion.** To enhance generalization and prevent overfitting, we employ a completion task where the full context is not provided to the hypernetwork (Figure 11). Specifically, we truncate the last 10%–30% of tokens from the context $\mathbf{c}$ to create a partial context $\mathbf{c}' \triangleq (c_1, \ldots, c_{N-k})$, with $k$ sampled uniformly between $0.1N$ and $0.3N$. The generated LoRA must infer the missing information to allow the LLM to recover the full original context $\mathbf{c}$. The objective is:

$$\mathcal{J}_{\mathrm{COMP}} = -\log P(\mathbf{c}\,|\,\texttt{<COMP>}; \Theta'_{\mathrm{GLoRA}}, \Theta_{\mathrm{LLM}}), \quad (11)$$

where $\Theta'_{\mathrm{GLoRA}} = f(\mathbf{c}'; \Theta_{\mathrm{T}}, \Theta_{\mathrm{M}}, \mathbf{m})$.

**Combination.** We train on both tasks jointly by randomly sampling between them. To improve training efficiency, short contexts are concatenated into longer sequences (see Appendix B.1 for details). The total objective is:

$$\mathcal{J}_{\mathrm{TOTAL}} = \lambda \mathcal{J}_{\mathrm{RECON}} + (1 - \lambda)\mathcal{J}_{\mathrm{COMP}}.$$

In practice, we set $\lambda = 0.5$.

### 4.2. Instruction Fine-Tuning

Following pretraining, the hypernetwork can effectively internalize contextual information. The Instruction Fine-Tuning (IFT) stage aligns this capability with downstream reasoning tasks, enabling the generation of adapters that support Question Answering (QA).

We utilize Context-Question-Answer $(\mathbf{c}, \mathbf{q}, \mathbf{a})$ datasets. In each training step (Figure 12), the hypernetwork generates

parameters $\Theta_{\mathrm{GLoRA}}$ based on the context $\mathbf{c}$. The question $\mathbf{q}$ and answer $\mathbf{a}$ are formatted into a chat template. We minimize the cross entropy loss of the answer tokens $\mathbf{a}$:

$$\mathcal{J}_{\mathrm{IFT}} = -\log P(\mathbf{a}\,|\,\mathbf{q}; \Theta_{\mathrm{GLoRA}}, \Theta_{\mathrm{LLM}}). \quad (12)$$

This ensures the generated LoRA not only stores the context but also understand and express it by answering questions.

## 5. Experiments

We employ Qwen3-8B (Yang et al., 2025) as the backbone language model. Unless otherwise specified, all experiments use a Meta LoRA rank of 128, a generated LoRA rank of 8, and $M = 148$ input memory embeddings according to Equation 4. The maximum context length is set to 1,150 tokens, and a 4-layer M2P Transformer is used. Experiments were conducted on a Linux server with eight NVIDIA A100 GPUs using the AdamW optimizer (Loshchilov & Hutter, 2019) with a linear scheduler and warmup.

### 5.1. Pretraining

We utilize a 6B token pretraining dataset from TransMLA (Meng et al., 2025), training for 1 epoch with a peak learning rate of 5e-5. Upon completion, we evaluate our hypernetwork SHINE's reconstruction and completion capabilities using articles of varying lengths from Wikitext-2 (Merity et al., 2017). The testing protocol mirrors pretraining: given a context, the hypernetwork generates LoRA weights. We then measure the loss and perplexity (PPL) of the LLM in reconstructing or completing the context solely via the generated LoRA, without direct access to the context.

As shown in Figure 5, both tasks achieve consistently low loss and PPL. A minor exception is observed at context lengths around 100 tokens; we attribute this to a subset of meaningless outliers in the dataset, as the median performance remains robust. Overall, these results demonstrate that SHINE effectively memorizes and completes texts.

### 5.2. Instruction Fine-Tuning

Following pretraining, we perform instruction fine-tuning. We categorize datasets into two types: **mqa** (multiple question–answer pairs per context) and **1qa** (single pair per context). Due to the scarcity of high-quality **mqa** datasets, we constructed **MS MARCO MQA**. We adopted contexts from MS MARCO (Nguyen et al., 2016) and utilized LLM to generate 15 question–answer pairs per context. Further details are provided in Appendix B.2.

#### 5.2.1. INSTRUCTION FINE-TUNING: MQA

We first fine-tune SHINE on a collection of **mqa** datasets, primarily comprising MS MARCO MQA (76%) alongside

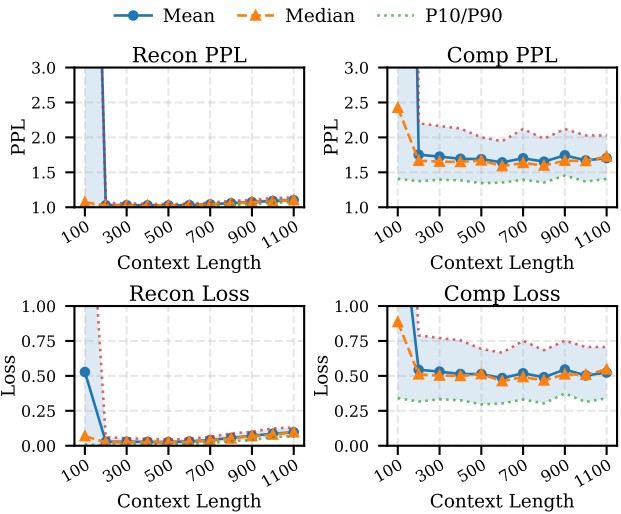

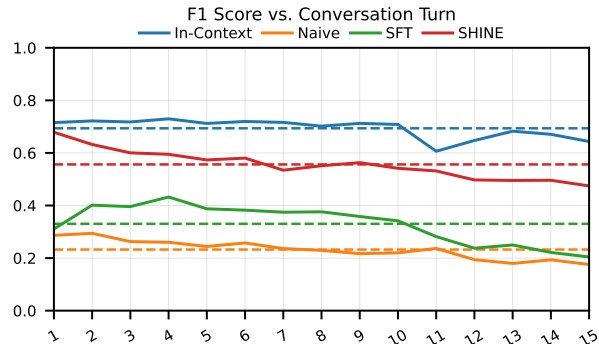

*Figure 5.* **Pretraining Results:** Reconstruction and completion loss/perplexity across varying context lengths. P10/P90 denote 10% quantile/90% quantile.

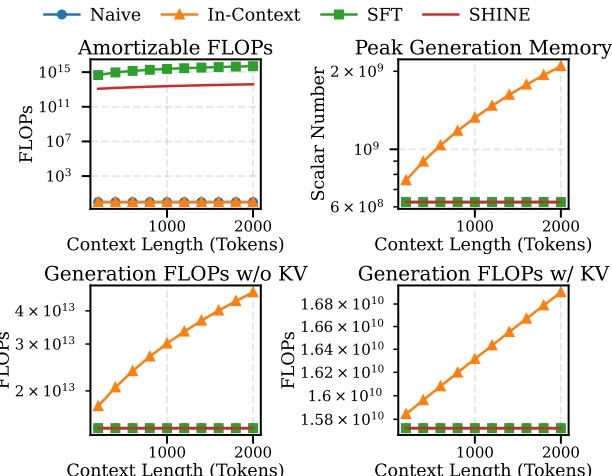

*Figure 7.* **Computation and Memory Costs:** Comparison of FLOPs and peak memory usage.

mance comparable to In-Context prompting, especially for 1-turn conversation, and substantially outperforms both SFT and the Naive baseline. We attribute the decline of SHINE on multi-turn conversations to lack of the time-consuming long-context post-training as in ICL models—in SHINE, although the original context has been absorbed in LoRA weights, the QA pairs keep increasing the conversation length, incurring long-context inference difficulties.

**Efficiency Analysis:** We compare time, computation, and memory costs. Table 1 details the time costs. *Amortizable time* refers to the one-time preparation cost (LoRA generating). Naive and In-Context have zero amortizable cost, whereas SFT incurs a high training overhead. SHINE requires only a single forward pass, resulting in negligible amortizable time (0.3s). For generation, SHINE matches the speed of Naive and SFT, avoiding the latency penalty of processing the original long context as in In-Context.

Figure 7 visualizes computation (FLOPs) and memory usage (see Appendix B.5 and B.6 for derivations). SHINE significantly reduces fine-tuning computation compared to SFT and lowers memory/compute costs during generation compared to In-Context methods.

*Table 1.* Comparison of Performance and Time Costs on MS MARCO MQA.

| METHOD | F1 SCORE | AMORTIZABLE TIME (S) | GENERATION TIME (S) |
|---|---|---|---|
| NAIVE | 23.2 | 0.0 | 11.0 |
| IN-CONTEXT | 69.4 | 0.0 | 14.2 |
| SFT | 33.0 | 29.3 | 11.0 |
| SHINE | 55.6 | 0.3 | 11.0 |

### 5.2.2. INSTRUCTION FINE-TUNING: 1QA

Following **mqa** tuning, we fine-tune on **1qa** datasets (listed in Appendix B.4) for 1 epoch at a learning rate of 1e-5. We

---

*Figure 6.* **Multi-Turn Conversation F1-Score:** Evaluation answer F1 scores using the MS MARCO MQA dataset, which contains 15 QA pairs per context.

open-source alternatives (see Appendix B.3). Training proceeds for 2 epochs with a peak learning rate of 3e-5.

**Evaluation Setup:** We evaluate the model by generating multi-turn conversations on the MS MARCO MQA test set. The LLM generates answers autoregressively, conditioning on the current question and previous generated answers. Crucially, the original context is hidden from the LLM; access to contextual information is mediated strictly through the generated LoRA weights.

**Baselines:** We compare SHINE against: (1) **Naive:** System prompt + conversation history (no context). (2) **In-Context:** System prompt + full context + conversation history, which is often the **golden baseline** due to full access to the context during inference. (3) **SFT:** We fine-tune a rank=8 LoRA (matching our generated rank) for 10 epochs on 5 conversations per context.

**Results:** As shown in Figure 6, SHINE achieves perfor-

*Table 2.* **1QA Performance (Answer F1 Score):** 1/2 MQA and 1/2 1QA denote models trained with half the MQA or 1QA data, respectively. The fully trained SHINE outperforms two variants on 7/9 datasets, verifying the effectiveness of both MQA and 1QA data. SQuAD-N follows the setting in (Chen et al., 2025) and refers to a dataset variant where contexts are concatenated to an average length of N tokens. The inclusion of distractor passages within these longer contexts serves to elevate task difficulty.

| | SINGLE-HOP QA | | | MULTI-HOP QA | | | SQUAD-N | | |
| METHOD | SQUAD | MS MARCO V1 | MS MARCO V2 | HOTPOT QA | MUSI-QUE | 2WIKI-MULTIHOPQA | SQUAD 512 | SQUAD 1K | SQUAD 2K |
|---|---|---|---|---|---|---|---|---|---|
| NAIVE | 22.0 | 19.6 | 16.0 | 26.9 | 11.8 | 27.8 | 22.0 | 22.0 | 22.0 |
| IN-CONTEXT | **86.8** | 34.2 | 31.3 | **68.7** | **36.3** | 48.7 | **85.9** | **85.1** | **84.9** |
| GEN ADAPTER | 70.3 | 35.0 | 27.9 | 40.8 | 19.4 | 32.9 | 48.8 | 43.0 | 39.9 |
| SHINE(1/2 MQA) | 59.3 | 39.1 | **40.8** | 57.7 | 28.2 | **61.0** | 49.2 | 42.7 | 33.8 |
| SHINE(1/2 1QA) | 62.2 | 40.2 | 39.9 | 56.2 | 26.7 | 57.8 | 50.9 | 42.4 | 34.6 |
| SHINE | 63.6 | **40.7** | 40.1 | 59.0 | 28.5 | 60.2 | 53.4 | 44.5 | 37.5 |

evaluate on six representative QA benchmarks: SQuAD (Rajpurkar et al., 2016), MS MARCO (v1/v2) (Nguyen et al., 2016), HotpotQA (Yang et al., 2018), MuSiQue (Trivedi et al., 2022), and 2WikiMultihopQA (Ho et al., 2020). For the last 3 datasets we use their default distractor setting. We excluded MS MARCO no answer data points from hypernetwork training and testing. F1 scores on test splits are reported where available (otherwise validation). We additionally include a hypernetwork baseline Generative Adapter (Chen et al., 2025).

Results are presented in Table 2. SHINE substantially outperforms the Naive baseline and achieves parity with, or occasionally exceeds, In-Context learning. It also significantly outperforms Generative Adapter in most cases. Notably, SHINE performs well on multi-hop QA tasks (HotpotQA, MuSiQue, 2Wiki) in the absence of explicit CoT steps. This capability suggests that the generated LoRA not only memorize the context, but also capture the underlying semantic structure in it, enabling reasoning over the parameterized context.

### 5.3. Comparison with Test-Time Training

We compare SHINE with leading Test-Time Training (TTT) baselines, such as SEAL (Zweiger et al., 2025) and PaST (Tang et al., 2026). TTT approaches typically enable continual learning through online parameter updates on test streams. SHINE similarly operates under the continual learning paradigm but bypasses the computational overhead of iterative optimization. It adapts by directly generating LoRA parameters by a hypernetwork in one forward pass, avoiding gradient-based training.

Standard TTT methods typically require optimization over multiple iterations or documents. Their performance varies by the number of training contexts ($n$). As shown in Table 3, compared to TTT methods with $n = 200$ (which generally yields the best results for them, balancing data sufficiency against catastrophic forgetting), SHINE outperforms these methods significantly. Crucially, while TTT requires iteratively training on hundreds of articles and using SFT and RL

techniques with synthetic data, SHINE generates the adapter parameters in a single forward pass with negligible computational overhead. Full comparisons for $n = \{1, 200, 2067\}$ are provided in Appendix B.7.

*Table 3.* **Comparison with Test-Time Training ($n = 200$) on SQuAD.** SHINE achieves superior performance with significantly lower latency.

| Method | Score |
|---|---|
| *Test Time Training (Full-FT, $n = 200$) — Iterative Optimization, long time, huge training overhead.* | |
| Base Model | 32.7 |
| Train on Passage | 36.0 |
| Train on Passage + Synthetic | 50.6 |
| Train on Passage + GPT-4.1 Synthetic | 59.4 |
| SEAL | 58.2 |
| PaST 50 | 58.9 |
| *Hypernetwork (One Forward Pass) — Instant Generation, negligible time, no training overhead.* | |
| SHINE | **63.6** |

### 5.4. Scalability Analysis

We evaluate the scalability of SHINE in two directions: scaling the backbone LLM and scaling the hypernetwork.

**Scaling with Backbone LLM.** We maintained the default settings for all hyperparameters (e.g., Generated LoRA rank, Meta LoRA rank, and M2P Transformer depth). We varied only the scale of the backbone LLM. All models were evaluated after completing 40% of the pretraining schedule. As illustrated in Figure 9,17,18, scaling LLM, M2P Transformer layer number, LoRA ranks all lead to better results.

**Scaling Hyperparameters.** We further investigated the impact of scaling the internal components of the hypernetwork. Table 4 presents the results with various configurations, all pretrained for 40% of the schedule. Increasing M2P Transformer layers, meta LoRA rank, and Generated LoRA rank all lead to PPL decrease, showing the scaling potential of

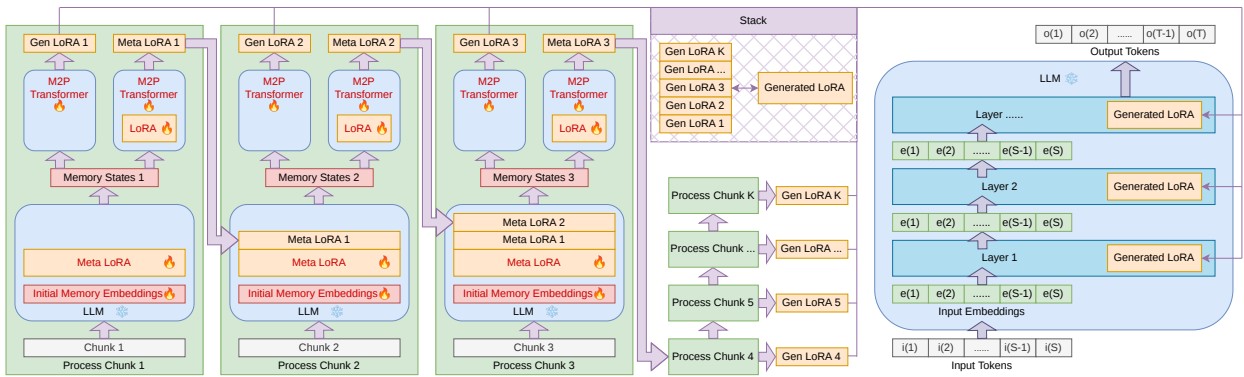

*Figure 8.* **SHINE-R**: A recurrent architecture for long context.

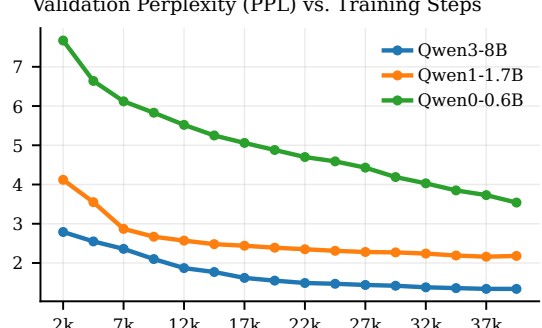

*Figure 9.* **Scaling with Backbone LLM:** Pretraining performance comparison across different sizes of backbone LLM.

our hypernetwork architecture.

*Table 4.* **Impact of Hyperparameter Scaling.** Perplexity (PPL) is reported on the validation set after 40% pretraining.

| M2P Transformer Layers | Meta LoRA Rank | Generated LoRA Rank | PPL |
|---|---|---|---|
| 4 | 4 | 4 | 1.88 |
| 6 | 4 | 4 | 1.79 |
| 4 | 8 | 8 | 1.63 |
| 4 | 32 | 8 | 1.57 |
| 4 | 128 | 4 | 1.50 |
| 4 | 128 | 8 | 1.32 |

### 5.5. Long Context

Inspired by RNNs, we design SHINE-R, where R denotes recurrent. As shown in Figure 8, SHINE-R splits a long context into chunks and processes them sequentially. For each chunk, M2P generates two LoRAs: one with a LoRA on the M2P Transformer and one without. The former is concatenated with the meta LoRA to update it for encoding later chunks, while the latter is saved. The saved LoRAs are concatenated at the end to form a LoRA encoding the full context. In theory, this can handle infinitely long contexts, with LoRA memory growing linearly with context length.

We continual pretrain and finetuning on long context up to 18K tokens, then evaluate on LongBench (Bai et al., 2024) QA and summarization tasks. As shown in Table 5, SHINE-R significantly outperforms Naive. Although it still trails In-Context, likely due to insufficient training under limited computational resources. SHINE-R greatly reduces inference VRAM cost in long-context settings and achieves much lower perplexity than origin SHINE, confirming the effectiveness of the recurrent architecture.

*Table 5.* Long context results on LongBench dataset.

| METHOD | 2WIKIMQA | HOTPOT QA | MULTIFIELDQA (EN) |
|---|---|---|---|
| NAÏVE | 26.7 | 27.0 | 11.7 |
| IN-CONTEXT | 45.5 | 55.9 | 49.7 |
| SHINE-R | 33.1 | 32.7 | 22.0 |

| METHOD | MUSIQUE | QASPER | QMSUM |
|---|---|---|---|
| NAÏVE | 11.62 | 14.02 | 4.76 |
| IN-CONTEXT | 31.66 | 42.38 | 22.44 |
| SHINE-R | 14.74 | 20.06 | 19.87 |

### 5.6. Ablation Studies

More ablation studies are in Appendix E.

## 6. Conclusion

We propose SHINE, a scalable in-context hypernetwork that can generate high quality LoRA parameters from diverse meaningful contexts in a single pass. By incorporating an in-context design and architectural innovations, SHINE has strong expressive power and uses a relatively small number of parameters. Trained end-to-end through a pretraining and instruction fine-tuning pipeline with 6 billion pretraining tokens, SHINE achieves strong performance across a range of complex tasks, surpassing existing baselines without showing signs of capacity saturation. Moreover, SHINE reduces computation overhead compared to other LLM adaptation approaches, demonstrating compelling potential for real-world deployment and further scaling.

## Acknowledgments

This work is supported by National Natural Science Foundation of China (62276003).

## Impact Statement

This paper presents work whose goal is to make LLM adaptation more efficient and accessible. There are many potential societal consequences of our work, none of which we feel must be specifically highlighted here.

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

# A. Architecture Details

## A.1. Post Layernorm

Transformer Layer has two designs based on the position of layernorm. One is pre-layernorm:

$$
\begin{aligned}
x' &= x + \text{Attention}\big(\text{LN}(x)\big) \\
y &= x' + \text{FFN}\big(\text{LN}(x')\big)
\end{aligned}
\tag{13}
$$

The other is post-layernorm:

$$
\begin{aligned}
x' &= \text{LN}\big(x + \text{Attention}(x)\big) \\
y &= \text{LN}\big(x' + \text{FFN}(x')\big)
\end{aligned}
\tag{14}
$$

Almost all modern LLMs use pre-layernorm, because it has an identity shortcut that allows gradients to flow directly through many layers, making very deep Transformers trainable without warmup tricks or careful tuning. In post-layernorm, gradients must pass through LayerNorm at every layer, which can cause gradient shrinkage and training divergence for deep models.

But, unlike modern LLMs, here we choose to use post-layernorm for our hypernetwork because identity shortcut is no longer needed. Hypernetwork is not deep, and it isn't a usual transformer. It maps memory hidden states into LoRA parameters, which naturally have huge distribution gaps between them. Under this situation, post-layernorm makes training more stable and converges much faster. Pre-layernorm is too sensitive to the input memory states and causes trouble in training.

## A.2. Reshape into LoRA

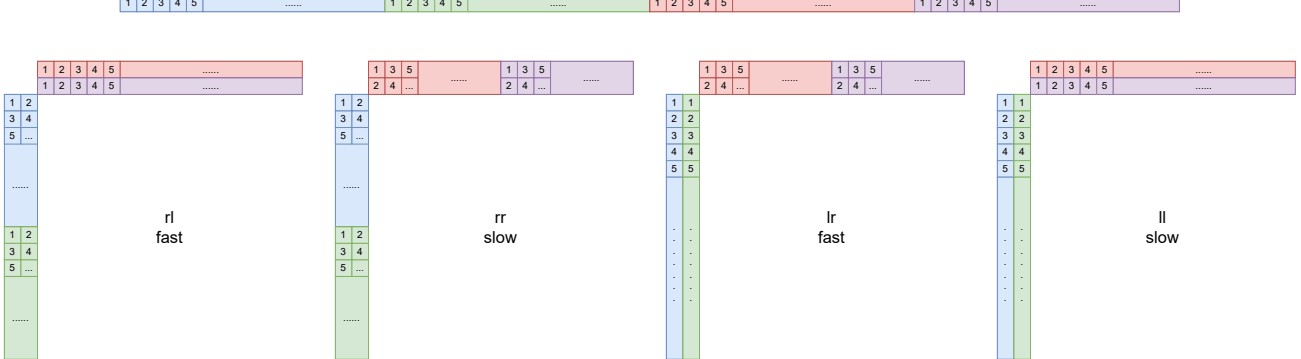

*Figure 10.* Reshape to LoRA: Different ways of transpose.

Define $\mathbf{A}^\top$ as the transpose of A. We have four kinds of transpose in the LoRA generating process:

$$
\text{rl:} \quad
\begin{aligned}
\mathbf{A} &= \text{Reshape}(\mathbf{T}[t : t + I \cdot r]) \in \mathbb{R}^{I \times r} \\
\mathbf{B} &= \text{Reshape}(\mathbf{T}[t + I \cdot r : t + I \cdot r + r \cdot R]) \in \mathbb{R}^{r \times O}
\end{aligned}
\tag{15}
$$

$$
\text{rr:} \quad
\begin{aligned}
\mathbf{A} &= \text{Reshape}(\mathbf{T}[t : t + I \cdot r]) \in \mathbb{R}^{I \times r} \\
\mathbf{B}^\top &= \text{Reshape}(\mathbf{T}[t + I \cdot r : t + I \cdot r + r \cdot R]) \in \mathbb{R}^{O \times r}
\end{aligned}
\tag{16}
$$

$$
\text{lr:} \quad
\begin{aligned}
\mathbf{A}^\top &= \text{Reshape}(\mathbf{T}[t : t + I \cdot r]) \in \mathbb{R}^{r \times I} \\
\mathbf{B}^\top &= \text{Reshape}(\mathbf{T}[t + I \cdot r : t + I \cdot r + r \cdot R]) \in \mathbb{R}^{O \times r}
\end{aligned}
\tag{17}
$$

$$
\text{ll:} \quad
\begin{aligned}
\mathbf{A}^\top &= \text{Reshape}(\mathbf{T}[t : t + I \cdot r]) \in \mathbb{R}^{r \times I} \\
\mathbf{B} &= \text{Reshape}(\mathbf{T}[t + I \cdot r : t + I \cdot r + r \cdot R]) \in \mathbb{R}^{r \times O}
\end{aligned}
\tag{18}
$$

In practice we find the rl and lr mode converges much faster than rr and ll mode. As visualized in Figure 10, we observed that rl and lr mode are more complex and mixed, while rr and ll are simply the outer product between two memory embeddings. And we think this may make it hard to optimize, so in practice we don't use them. We do all our experiments using the rl mode.

# B. Training and Experiments

### B.1. Pretraining: Concatenating Short Contexts into Longer Ones

During pretraining, we set a maximum sequence length and attempt to concatenate short contexts as much as possible, provided their combined length does not exceed the maximum sequence length.

For example, consider three contexts, $A$, $B$, and $C$. The goal is to concatenate these contexts into a single longer context, which we denote as $D$. The concatenated context takes the form $D = A < \text{EOT} > B < \text{EOT} > C$, where $< \text{EOT} >$ is a predefined special token used to separate distinct contexts. Here each of A, B and C is randomly assigned to either a reconstruction or completion task. If a context is designated for the completion task, it is truncated.

During training, the sequence of conversations might look as follows: `<USR> <RECON> <ASSISTANT>` G `<USR> <COMP> <ASSISTANT>` H `<USR> <RECON> <ASSISTANT>` I. Here, $G$, $H$, and $I$ are random permutations of $A$, $B$, and $C$, and the task type (either `<RECON>` or `<COMP>`) is determined based on the assigned task. By using random permutation rather than one-to-one alignment, we aim to enhance the model's generality. This approach ensures that any position with the generated LoRA can attend to any position in the given context.

### B.2. MS MARCO MQA

Due to the lack of large-scale, high-quality open-source datasets for multi question-answering (MQA) based on one context, we construct **MS MARCO MQA**, a new dataset derived from the MS MARCO corpus.

Each data point in MS MARCO MQA consists of a *context* and a *conversation*. The context is obtained from the original MS MARCO dataset: for each data point, we concatenate all passages in it using \n\n to form a single unified context.

Given this context, we generate 15 question–answer (QA) pairs using **Qwen-Flash**. Among these, the first 10 are *specific* questions targeting concrete facts, entities, or details explicitly stated in the context, while the remaining 5 are *general* questions focusing on the main idea, purpose, or high-level summary of the context. QA generation is guided by the following prompt:

```
PROMPT_TEMPLATE = (
    "You are given a context below. Your task is to generate 15 diverse questions and
    answers based on this context:\n\n"
    "- 10 should be **specific questions** that ask for concrete details, facts, or
    entities explicitly mentioned in the context.\n"
    "- 5 should be **general questions** that ask about the main idea, purpose, or a high-
    level summary of the context.\n\n"
    "Each answer must be directly extractable from the context (i.e., an exact span or
    minor paraphrase for fluency). Do not invent information.\n\n"
    "Format your response as a JSON list of 15 objects, each with \"question\" and \"
    answer\" keys, like:\n"
    "[\n"
    "  {\"question\": \"...\", \"answer\": \"...\"},\n"
    "  {\"question\": \"...\", \"answer\": \"...\"},\n"
    "  ...\n"
    "]\n\n"
    "Context:\n"
    "{context_placeholder}"
)
```

After generation, we apply a two-stage validation process. First, we verify that the model output strictly follows the required JSON format. Second, we validate the factual correctness of the generated QA pairs by re-invoking Qwen-Flash as an automatic fact-checker, using the following validation prompt:

```
    VALIDATION_PROMPT = """You are an expert fact-checker. Given a context and a list of
    15 question-answer pairs, determine if ALL answers are:
    1. Fully grounded in the context -- meaning the answer is either:
       - An exact substring of the context, OR
       - A minor, fluent paraphrase that does not add, remove, or distort any factual
    detail (e.g., changing 'was founded in 1976' to 'founded in 1976' is OK; saying '
    started in the 70s' is NOT OK).
```

```
2. Factually consistent with the context.
3. Paired with a clear, relevant question that can be answered from the context.

If ANY answer fails these criteria, respond with:
{{"valid": false, "reason": "Brief reason"}}

If ALL are valid, respond with:
{{"valid": true}}

Context:
{context}

QA Pairs:
{qa_list_str}
"""
```

Any data point that fails either the format or validation check needs to be regenerated. If one fails too many times or has inappropriate contexts, it will be discarded.

We initially generate MS MARCO MQA using all data points from **MS MARCO V1**, preserving the original train/validation/test splits. However, we found that the resulting training set was insufficient for large-scale training. To address this, we further augment the train split of the dataset by sampling approximately one-third of the train data points from **MS MARCO V2** and applying the same generation and validation pipeline.

The final dataset consists of 366K training examples (82K from MS MARCO V1 and 284K from MS MARCO V2), along with 10K validation and 10K test examples.

### B.3. MQA Collection Datasets

Our datasets are processed from MS MARCO MQA, QUAC: (Choi et al., 2018), COQA: (Reddy et al., 2019), DROP: (Dua et al., 2019), NarrativeQA: (Kociský et al., 2018), QuAIL: (Rogers et al., 2020; Xiao et al., 2024; Enevoldsen et al., 2025; Muennighoff et al., 2022), PwC: (Ge et al., 2024), SQuAD: (Rajpurkar et al., 2016), RACE: (Lai et al., 2017), NewsQA: (Trischler et al., 2017), DuoRC: (Saha et al., 2018).

Due to the lack of qualified open source **mqa** datasets, we generate MS MARCO MQA by ourselves and it accounts for 76% of the total data volume. Some datasets here are naturally **mqa**. Some other datasets here are originally **1qa** but the same contexts appear multiple times. We simply gather question-answer pairs of the same context together to turn them into **mqa**.

### B.4. 1QA Collection Datasets

Our 1QA datasets include MS MARCO: (Nguyen et al., 2016), SQuAD: (Rajpurkar et al., 2016), RACE: (Lai et al., 2017), DuoRC: (Saha et al., 2018), HotpotQA: (Yang et al., 2018), MuSiQue: (Trivedi et al., 2022), 2WikiMultihopQA: (Ho et al., 2020). All datasets naturally have one context and a question-answer pair per data point.

### B.5. Computation

We report floating-point operations (FLOPs) using the standard convention that one multiply–accumulate (MAC) corresponds to two FLOPs (one multiplication and one addition), i.e., $1 \text{ MAC} = 2 \text{ FLOPs}$.

#### B.5.1. QWEN3 TRANSFORMER ARCHITECTURE

We consider a Transformer with hidden size $H$, context length $N$ (number of tokens), $L$ layers, and vocabulary size $V$. Each layer uses grouped-query attention (GQA), where the key/value projection width is $H/4$, and a SwiGLU-style MLP with intermediate dimension $3H$.

**FLOPs without KV cache.** We first account for the Transformer blocks and then separately include the final vocabulary projection. A dense linear map from $\mathbb{R}^{d_{\text{in}}}$ to $\mathbb{R}^{d_{\text{out}}}$ applied to $M$ tokens costs $2Md_{\text{in}}d_{\text{out}}$ FLOPs.

*Attention projections (per layer).*

$$\text{FLOPs}_Q = 2NH^2, \quad \text{FLOPs}_K = 2NH \cdot \tfrac{H}{4} = \tfrac{1}{2}NH^2, \quad \text{FLOPs}_V = \tfrac{1}{2}NH^2, \quad \text{FLOPs}_O = 2NH^2, \quad (19)$$

which sum to $5NH^2$ FLOPs per layer.

*Self-attention matmuls (per layer).* Full self-attention over $N$ tokens incurs $4N^2H$ FLOPs per layer from $QK^\top$ and $AV$.

*MLP (per layer).* The SwiGLU MLP (`gate`, `up`, `down`) with intermediate size $3H$ costs $18NH^2$ FLOPs per layer.

Combining these terms, the forward-pass FLOPs of the Transformer blocks (excluding the output layer) are

$$\text{FLOPs}_{\text{blocks}}^{(\text{no KV})} = L\big(23NH^2 + 4N^2H\big). \quad (20)$$

*Final vocabulary projection.* In standard language-model inference and training, the vocabulary projection $\mathbb{R}^H \to \mathbb{R}^V$ is applied only to the final hidden state corresponding to the last token. Its cost is therefore

$$\text{FLOPs}_{\text{vocab}} = 2HV, \quad (21)$$

independent of the sequence length $N$.

Thus, the total forward-pass FLOPs without KV cache are

$$\text{FLOPs}_{\text{total}}^{(\text{no KV})} = L\big(23NH^2 + 4N^2H\big) + 2HV. \quad (22)$$

**FLOPs with KV cache.** With a KV cache, generation proceeds one token at a time. At each decoding step, only the newly generated token is processed through the Transformer layers, while keys and values from the previous $N$ tokens are reused.

*Attention projections (per layer, one token).*

$$\text{FLOPs}_Q = 2H^2, \quad \text{FLOPs}_K = 2H \cdot \tfrac{H}{4} = \tfrac{1}{2}H^2, \quad \text{FLOPs}_V = \tfrac{1}{2}H^2, \quad \text{FLOPs}_O = 2H^2, \quad (23)$$

which sum to $5H^2$ FLOPs per layer.

*Attention matmuls (per layer, one token attending to $N$ cached tokens).*

$$QK^\top : (1 \times H)(H \times N) \Rightarrow 2NH, \qquad AV : (1 \times N)(N \times H) \Rightarrow 2NH, \quad (24)$$

for a total of $4NH$ FLOPs per layer.

*MLP (per layer, one token).* The SwiGLU MLP with intermediate size $3H$ costs $18H^2$ FLOPs per layer.

Combining these terms, the per-step decoding FLOPs of the Transformer blocks are

$$\text{FLOPs}_{\text{blocks}}^{(\text{KV})}(N) = L\big(23H^2 + 4NH\big). \quad (25)$$

*Final vocabulary projection.* As in the no-cache case, the vocabulary projection is applied only to the newly generated token, incurring

$$\text{FLOPs}_{\text{vocab}} = 2HV. \quad (26)$$

Hence, the total FLOPs for one autoregressive decoding step at context length $N$ are

$$\text{FLOPs}_{\text{total}}^{(\text{KV})}(N) = L\big(23H^2 + 4NH\big) + 2HV. \quad (27)$$

### B.5.2. HYPERNETWORK TRANSFORMER ARCHITECTURE

We consider a Transformer with hidden size $H$, context length $N$, $L$ layers. Each layer uses full attention (key/value projection width $H$) and a SwiGLU-style MLP with intermediate dimension $2H$.

*Attention projections (per layer).*

$$\text{FLOPs}_Q = 2NH^2, \quad \text{FLOPs}_K = 2NH^2, \quad \text{FLOPs}_V = 2NH^2, \quad \text{FLOPs}_O = 2NH^2, \quad (28)$$

which sum to $8NH^2$ FLOPs per layer.

*Self-attention matmuls (per layer).* Full self-attention over $N$ tokens incurs $4N^2H$ FLOPs per layer.

*MLP (per layer).* The SwiGLU MLP with intermediate size $2H$ costs $12NH^2$ FLOPs per layer.

Thus, the forward-pass FLOPs of the Transformer blocks (excluding the output layer) are

$$\text{FLOPs}_{\text{blocks}}^{(\text{no KV})} = L\big(20NH^2 + 4N^2H\big). \tag{29}$$

### B.5.3. AMORTIZABLE FLOPS

Denote $M = \frac{37}{2}r$ as the number of memory tokens, $C$ as the number of context tokens, $L$ as the layer number of LLM, and $L'$ as the layer num of hypernetwork.

**SHINE.** Amortizable FLOPs are the process of the forward pass to generate LoRA.

Generate memory state:

$$\text{FLOPs}_1 = L\big(23(C + M)H^2 + 4(C + M)^2H\big). \tag{30}$$

Memory state to parameters:

$$\text{FLOPs}_2 = \frac{1}{2}L'M\big(20LH^2 + 4L^2H\big) + \frac{1}{2}L'L\big(20MH^2 + 4M^2H\big) \tag{31}$$

The final FLOPs is:

$$\text{FLOPs}_{\text{SHINE}} = \text{FLOPs}_1 + \text{FLOPs}_2 = LH\big(23(C + M)H + 4(C + M)^2 + 20ML'H + 2LL'M + 2LM^2\big) \tag{32}$$

**SFT.** During fine-tuning, we use full-sequence forward (no KV cache) and compute the LM loss over all $C$ token positions. Hence, the total forward-pass FLOPs are

$$\text{FLOPs}_{\text{total}}^{(\text{fwd})} = L\big(23CH^2 + 4C^2H\big) + 2CHV. \tag{33}$$

*Training FLOPs (forward + backward).* Using the standard approximation that backward propagation costs roughly twice the forward computation, we estimate the fine-tuning FLOPs as

$$\text{FLOPs}_{\text{total}}^{(\text{ft})} \approx 3\,\text{FLOPs}_{\text{total}}^{(\text{fwd})} \approx 3\Big[L\big(23CH^2 + 4C^2H\big) + 2CHV\Big]. \tag{34}$$

Assume fine-tuning for $T$ iterations, the SFT FLOPs is:

$$\text{FLOPs}_{\text{SFT}} \approx 3T\Big[L\big(23CH^2 + 4C^2H\big) + 2CHV\Big]. \tag{35}$$

**Naive** and **In-Context** have no amortizable FLOPs.

### B.5.4. GENERATION FLOPS

Further denote $I$ as the number of input tokens (prompt + conversation history + question, context not included).

**SHINE, SFT and Naive** are the same. Without KV cache, the FLOPs is:

$$\text{FLOPs}_{\textbf{SHINE}}^{\text{noKV}} = \text{FLOPs}_{\textbf{SFT}}^{\text{noKV}} = \text{FLOPs}_{\textbf{Naive}}^{\text{noKV}} = L\big(23IH^2 + 4I^2H\big) + 2HV. \tag{36}$$

With KV cache, the FLOPs is:

$$\text{FLOPs}_{\text{SHINE}}^{\text{withKV}} = \text{FLOPs}_{\text{SFT}}^{\text{withKV}} = \text{FLOPs}_{\text{Naive}}^{\text{withKV}} = L\big(23H^2 + 4IH\big) + 2HV. \tag{37}$$

**In-Context** takes extra $C$ context tokens as input. Without KV cache, the FLOPs is:

$$\text{FLOPs}_{\textbf{InContext}}^{\text{noKV}} = L\big(23(I + C)H^2 + 4(I + C)^2H\big) + 2HV. \tag{38}$$

With KV cache, the FLOPs is:

$$\text{FLOPs}_{\textbf{InContext}}^{\text{withKV}} = L\big(23H^2 + 4(I + C)H\big) + 2HV. \tag{39}$$

**B.6. Memory**

We estimate the *peak extra memory* beyond model parameters, defined as the maximum number of floating-point scalars that must be resident simultaneously during computation. All results are reported in terms of scalar counts; multiplying by the bytes per scalar $b$ (e.g., $b=2$ for FP16/BF16, $b=4$ for FP32) yields memory usage in bytes.

We consider two execution regimes: (i) full-sequence computation without a KV cache, and (ii) autoregressive decoding with a KV cache. For each regime, we distinguish between *standard attention*, which materializes the full $N \times N$ attention matrix, and *memory-efficient attention* (e.g., FlashAttention-style), which avoids storing quadratic-size attention tensors at peak.

Throughout this analysis, we assume the **Qwen3 Transformer architecture** with grouped-query attention (GQA), key/value width $H/4$, and MLP hidden width $3H$.

**No KV cache.** *Peak extra memory under memory-efficient attention.* Under memory-efficient attention, the peak extra memory per layer is dominated by activation storage rather than attention scores. In particular, the largest resident tensors are:

1. the input hidden states of size $N \times H$, and

2. the MLP intermediate activations of size $N \times 3H$,

along with additional $O(NH)$ buffers for attention outputs, residual connections, and layer normalization.

Consequently, the peak extra memory across all layers scales as

$$\text{Mem}_{\text{peak}}^{\text{(no KV)}} \approx c\,LNH, \tag{40}$$

where $c$ is a modest architecture-dependent constant (typically in the range 4–6 in practice).

If we retain only the dominant activation tensors (hidden states and MLP intermediates), a simple closed-form approximation is

$$\text{Mem}_{\text{peak}}^{\text{(no KV)}} \approx L\,(NH + 3NH) = 4LNH \quad \text{(scalars)}. \tag{41}$$

*Peak extra memory under standard attention.* When standard attention is used, the attention matrix $A \in \mathbb{R}^{N \times N}$ must be materialized for each layer. In addition to the $O(LNH)$ activation memory described above, this contributes an additional quadratic term:

$$\text{Mem}_{\text{peak}}^{\text{(no KV, std-attn)}} \approx 4LNH + LN^2 \quad \text{(scalars)}. \tag{42}$$

The precise coefficient of the $LN^2$ term depends on whether raw attention scores, softmax-normalized probabilities, or both are stored, but the asymptotic scaling remains quadratic in $N$.

**With KV cache.** During autoregressive decoding, the dominant source of extra memory at long context lengths is the key–value (KV) cache. For each layer, the cache stores keys and values of width $H/4$ for all previously processed tokens. The per-layer cache size is therefore

$$\text{Mem}_{\text{cache, per layer}}^{\text{(KV)}} = N \cdot \tfrac{H}{4} + N \cdot \tfrac{H}{4} = \tfrac{1}{2}NH \quad \text{(scalars)}. \tag{43}$$

Aggregating across $L$ layers yields a total KV cache memory of

$$\text{Mem}_{\text{cache, total}}^{\text{(KV)}} = \tfrac{1}{2}LNH \quad \text{(scalars)}. \tag{44}$$

The additional *working memory* required to compute a single new token during decoding scales as $O(LH)$ and is typically negligible compared to the KV cache when $N$ is large. As a result, the peak extra memory during decoding is well-approximated by the cache term alone.

For **SHINE**, **SFT**, and **Naive**, the effective sequence length is $N = I$, whereas for **In-Context**, $N = I + C$.

## B.7. Compare with Test Time Training

Table 6 is the full results from paper (Tang et al., 2026). They combine SFT or RL techniques and train on Qwen-2.5-7B-Instruct, and may even use test time synthetic data for better training. But in the end, all of them are defeated by SHINE by a large margin. SHINE not only gets better results, but also greatly reduces the training overhead and time cost.

*Table 6.* Prior test time training results on SQuAD

| Method | Single Passage | CPT ($n = 200$; full-FT) | CPT ($n = 2067$; full-FT) |
|---|---|---|---|
| Base Model | 32.7 | 32.7 | 29.0 |
| Train on Passage | 33.5 | 36.0 | 31.2 |
| Train on Passage + Synthetic | 39.7 | 50.6 | 43.4 |
| Train on Passage + GPT-4.1 Synthetic | 46.3 | 59.4 | 49.2 |
| SEAL | 47.0 | 58.2 | 46.4 |
| PaST 50 (Ours) | 50.8 (+11.1) | 58.9 (+8.3) | 47.4 (+4.0) |
| PaST 50 × 2 (Ours) | 56.9 (+17.2) | 58.7 (+8.1) | 49.2 (+5.8) |

*Table 7.* SHINE result on SQuAD

| Method | Hypernetwork (one single forward pass) |
|---|---|
| **SHINE** | **63.6** |

# C. Hypernetwork Architecture Analysis

In this section, we provide a comparative analysis of our proposed method against existing hypernetwork architectures. We evaluate these approaches based on three distinct metrics: **Parameter Efficiency**, **Bottleneck Dimension**, and **Expressive Power**.

## C.1. Preliminaries

Consider a base Large Language Model (LLM) with $L$ layers, a hidden state dimension $H$, and a total parameter count $P$. In our experimental setup utilizing Qwen3-8B, we define $L = 36$ and $H = 4096$.

## C.2. Proposed Architecture (Ours)

Our architecture employs a Transformer-based hypernetwork designed to maximize expressive power through global attention mechanisms. Let $r$ denote the Meta-LoRA rank and $L'$ the number of hypernetwork layers. The total parameter count of our hypernetwork is estimated as:

$$P_{\text{ours}} \approx (\frac{L'}{L} + \frac{2r}{H})P \tag{45}$$

In practice, we set $L' = 4$ and $r = 128$, achieving high parameter efficiency. Crucially, our design imposes **no structural bottleneck**; the memory states maintain the same dimensionality as the final output LoRA weights.

Regarding expressive power, the architecture benefits significantly from the self-attention mechanism, which enables every memory token to attend to any other token. This facilitates deep information exchange and the modeling of complex dependencies between parameters across different layers.

## C.3. Comparison with Generative Adapter

The Generative Adapter (Chen et al., 2025) generates weights utilizing distinct projection matrices $A_1 \in \mathbb{R}^{d_{\text{out}} \times d_r}$, $A_2 \in \mathbb{R}^{d_r \times d_h}$, $B_1 \in \mathbb{R}^{d_h \times d_r}$, and $B_2 \in \mathbb{R}^{d_r \times d_{\text{out}}}$ for each target parameter $W \in \mathbb{R}^{d_{\text{out}} \times d_{\text{in}}}$. The parameter complexity for this hypernetwork is given by:

$$P_{\text{gen}} \approx (d_{\text{out}} + d_{\text{in}}) \times d_r \times 4 \tag{46}$$

With a standard rank setting of $d_r = 1024$, generating LoRA weights for all modules would result in a hypernetwork size approximating that of the base LLM ($P_{\text{gen}} \approx P$), which is computationally prohibitive. Consequently, this method is practically constrained to generating weights solely for the output projection layer of the attention mechanism.

Furthermore, its expressive power is limited by its additive nature. It performs an outer product of each hidden state with itself, summing them at each layer before transforming them into LoRA weights via linear transformations and SVD normalization. Unlike our Transformer-based approach, the parameters in the Generative Adapter cannot exchange information dynamically, limiting the global coherence of the generated weights.

In summary, while the Generative Adapter shares the advantage of having no bottleneck, it lacks parameter efficiency and expressive power compared to our method.

## C.4. Comparison with ICAE

ICAE (Ge et al., 2024) can be characterized as an in-context gist-based approach combined with Meta LoRA. In practice, it utilizes a Meta-LoRA rank of 128, identical to ours. However, as it lacks the M2P Transformer component, its parameter count is slightly lower and can be estimated as:

$$P_{\text{ICAE}} \approx \frac{2r}{H}P \tag{47}$$

Despite this, ICAE suffers from restrictive bottlenecks. The only information passed by the context is encoded in the gist tokens, resulting in a bottleneck of $H \times T$, where $T$ represents the number of gist tokens. Consequently, the context length is effectively limited; typically, it should not exceed three times the number of gist tokens to avoid a rapid increase in loss.

Compared to our approach, ICAE possesses fewer parameters but is severely constrained by the gist token bottleneck, which negatively impacts its expressive power. Additionally, ICAE introduces an inference latency overhead due to the gist tokens, whereas our method introduces zero overhead during inference.

### C.5. Comparison with Compositional Adapter

The Compositional Adapter (Jukic et al., 2025) also operates within an in-context setting. However, it directly utilizes the mean of the output tokens from the last LLM layer, mapping it via a linear layer to a highly compressed rank $r$, and subsequently projecting this to the full LoRA dimensions. While this results in a minimal parameter count, it introduces a severe information bottleneck due to the small $r$. Furthermore, it typically does not follow a pretraining and instruction fine-tuning paradigm like ours; instead, it is trained directly on specific tasks with significantly less data.

Compared to our architecture, the Compositional Adapter is far more parameter-efficient but suffers from a critical bottleneck and limited expressive power. It is restricted to specific fine-tuning tasks, whereas our method is capable of converting arbitrary context into LoRA weights. While easier to train, it is better suited for smaller-scale tasks rather than complex knowledge injection.

### C.6. Comparison with Text-to-LoRA

Text-to-LoRA (Charakorn et al., 2025) is a prominent work in hypernetwork-based LoRA generation, but it addresses a fundamentally different task. Their hypernetwork maps task descriptions to LoRAs, whereas ours maps entire contexts to LoRAs. From a task perspective, our objective is significantly more challenging: while their goal is to alter the style of the LLM or elicit latent behaviors, our task requires injecting specific contextual knowledge into the LLM—knowledge that may be entirely new or contradictory to the model's priors. Consequently, our generated LoRAs must be more powerful and are inherently harder to train.

Architecturally, Text-to-LoRA reuses a Multi-Layer Perceptron (MLP) multiple times and concatenates the outputs to form a full LoRA. While their parameter count is much lower than ours, reusing an MLP introduces a severe bottleneck due to the low dimensionality of the MLP input. Furthermore, MLPs inherently lack the expressive power of Transformers.

In conclusion, direct comparison is difficult due to the divergence in task objectives. While their architecture may be effective for their tasks, its limited expressive power renders it unsuitable for the knowledge injection tasks we address.

### C.7. Comparison with Drag-and-Drop LLMs

Drag-and-Drop LLMs (DnD) (Liang et al., 2025) also uses a hypernetwork to generate LoRA weights, but it targets a different problem setting. DnD learns a mapping from *prompts* to LoRA adapters, whereas our method maps *entire contexts* (i.e., substantially richer conditioning signals) to LoRA adapters. In this sense, DnD is similar to prompt-conditioned adapter generation approaches such as Text-to-LoRA (Charakorn et al., 2025), while our setting is strictly more challenging due to the broader and more informative input it must encode.

From an architectural perspective, DnD draws inspiration from recurrent diffusion (Wang et al., 2025) by representing weight information as tokens and uses a convolutional decoder as the hypernetwork. Similar to Text-to-LoRA (Charakorn et al., 2025), this design lacks expressive power and exhibits limited scalability relative to our method, but it remains a valuable contribution and may be well suited to its intended task.

In conclusion, direct comparison is difficult due to the divergence in task objectives. While their architecture may be effective for their tasks, its limited expressive power renders it unsuitable for the knowledge injection tasks we address.

### C.8. Conclusion

Our proposed architecture demonstrates strong expressive power while maintaining a relatively low parameter count. We believe that, compared to prior works, it offers superior potential for scaling and practical deployment in complex scenarios.

# D. Visualization

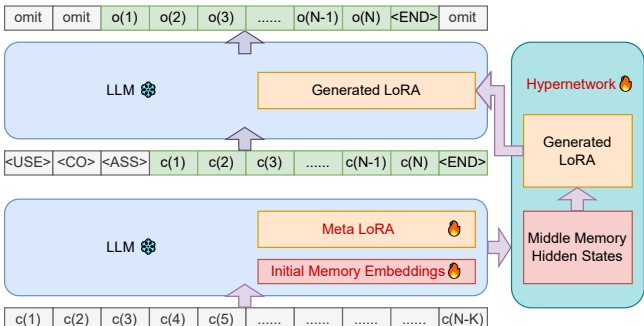

*Figure 11.* **Completion Task:** The hypernetwork encodes a truncated context into a LoRA. The LLM must utilize this LoRA to recover original context and complete the missing part.

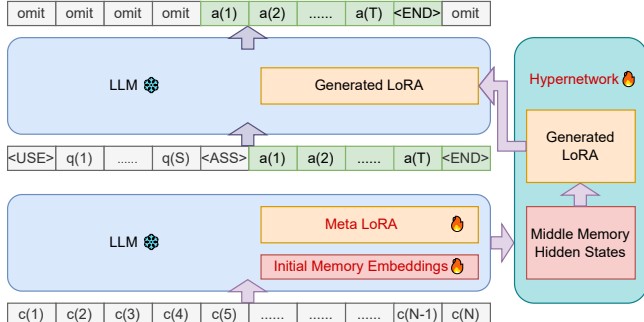

*Figure 12.* **Instruction Fine-Tuning:** The hypernetwork encodes context into LoRA. The model is then prompted to answer questions based on the context

# E. More Ablation Studies

## E.1. Architecture Design: The Bitter Lesson

Our architecture design is natural and well motivated. We adopt an in-context design to make the hypernetwork sufficiently expressive while keeping the number of parameters small, which requires reusing the parameters of the LLM. We introduce the Layer & Token Transformer because applying full attention over all tokens would incur excessive computational overhead.

However, the current design wastes many prior structural knowledges. For example, we do not exploit prior structural knowledge from LoRA. Given memory states $\mathbf{M} \in \mathbb{R}^{B \times L \times M \times H}$, we treat all $L \times M$ memory embeddings in the same way. In practice, these embeddings are not homogeneous: some are reshaped into the same LoRA module, while others are reshaped into LoRA modules that are far apart.

To address this, we introduce *coupled cross-attention layers*. A coupled cross-attention layer is similar to the original token attention layer (each token attends to all tokens in the same layer), except that it uses masked attention rather than full attention. We precompute whether each memory token will be reshaped into the $A$ or $B$ matrix of a specific LoRA. During cross-attention, a token corresponding to the $A$ matrix of a LoRA can only attend to tokens corresponding to the $B$ matrix of the same LoRA, and vice versa. The motivation is that the $A$ and $B$ matrices within the same LoRA are more correlated and should communicate more frequently.

We experiment with several configurations, including four coupled cross-attention layers, two original layers plus two coupled layers, and three original layers plus one coupled layer. As shown in Figure 13, couple layers do help the loss drop faster at the beginning. However, as training proceeds, the loss decreases much more slowly, and later all performance becomes worse than that of the original model with four full-attention layers.

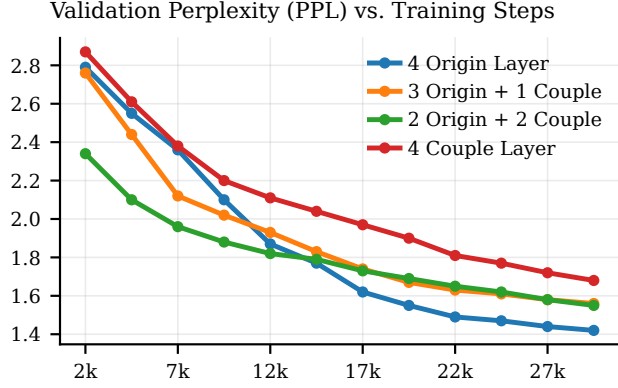

*Figure 13.* **Pretraining Validation PPL that Reflects the Bitter Lesson**

As the Bitter Lesson suggests, *"general methods that rely on computation and data ultimately outperform methods that rely on human-designed priors or domain-specific knowledge, even if the latter show faster early progress"*. Although the design of our hypernetwork does not explicitly incorporate many available priors, we believe it is already sufficiently powerful and not easily improved by explicitly adding such prior knowledge.

## E.2. Ablation Study on M2PTransformer Architecture Design

We perform ablation studies comparing our M2P Transformer with alternative architecture designs to demonstrate the effectiveness of our proposed design.

- **Linear small**: A two-layer linear network with hidden size 8192.

- **Linear**: An eight-layer linear network with hidden size 8192, containing nearly the same number of parameters as the M2P Transformer.

- **Linear Residual**: The Linear architecture with residual connections.

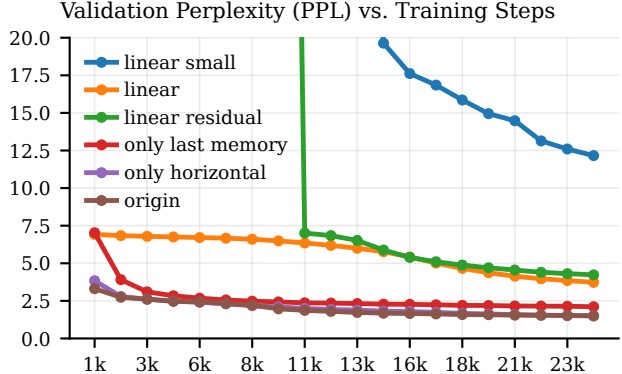

*Figure 14.* **Pretraining PPL for various alternatives of M2P Transformers**

- **Only Last Memory**: Instead of gathering memory states from all layers, this variant uses only the last-layer memory states and replicates them across all layers.

- **Only Horizontal**: An M2P Transformer variant with a four-token transformer.

- **Origin**: The original M2P Transformer with a two-token transformer and a two-layer transformer.

The results show that Transformer-based architectures generally perform much better than linear projection architectures. Using only the last-layer memory states performs worse than using memory states from all layers, demonstrating that aggregating memory states across layers is necessary. Although Origin and Only Horizontal achieve similar performance in the later stages, Origin converges faster during the early stages.

### E.3. More SFT

Tables 8, 9, and 10 report the F1-score and runtime of LoRA, DoRA (Liu et al., 2024), and IA3 (Liu et al., 2022) on the test set of MS MARCO MQA. Each entry is formatted as **F1-score(Time)**, where the runtime is measured in seconds.

When num_sample equals $n$, we generate $n + 1$ conversations for the same context in the same way. Among them, $n$ conversations are used for fine-tuning, and one conversation is used for testing. This corresponds to a very strong supervised fine-tuning (SFT) setting. The top row shows the number of fine-tuning epochs.

SHINE achieves an F1-score of $55.6$ with only $0.3$ seconds, requiring less time than fine-tuning a single sample for one epoch.

*Table 8.* LoRA results on MS MARCO MQA. Each entry denotes F1-score(Time), where time is measured in seconds.

| num_sample \ num_epoch | 5 | 10 | 20 |
|---|---|---|---|
| 3 | 24.56 (14.0) | 27.78 (28.1) | 35.05 (56.1) |
| 5 | 25.75 (23.40) | 32.72 (46.8) | 51.97 (93.5) |
| 10 | 32.81 (46.8) | 50.63 (93.5) | 58.50 (187.0) |

### E.4. Relationship between Context Length, Text Quality, and Hypernetwork Results

We visualize the relationship between text quality, text length, and answer F1-score. Here, text quality is measured by calculating perplexity (PPL) on Qwen3-8B-Base, a pretrained version of Qwen3-8B.

**Text Quality vs. F1-Score.** The higher the text quality is, the better it can be converted into a hypernetwork.

**Text Length vs. F1-Score.** The shorter the text is, the better it can be converted into a hypernetwork.

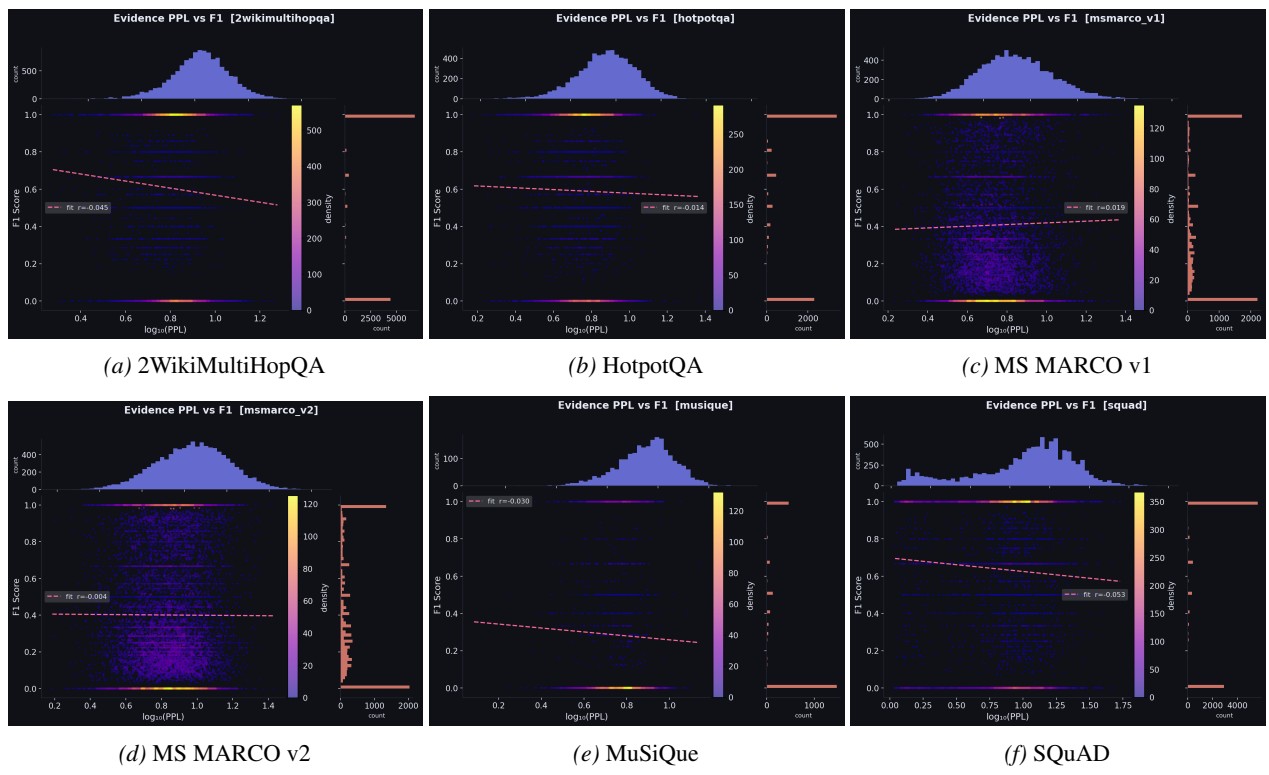

*(a)* 2WikiMultiHopQA        *(b)* HotpotQA        *(c)* MS MARCO v1

*(d)* MS MARCO v2        *(e)* MuSiQue        *(f)* SQuAD

*Figure 15.* Relationship between text quality measured by PPL and answer F1-score.

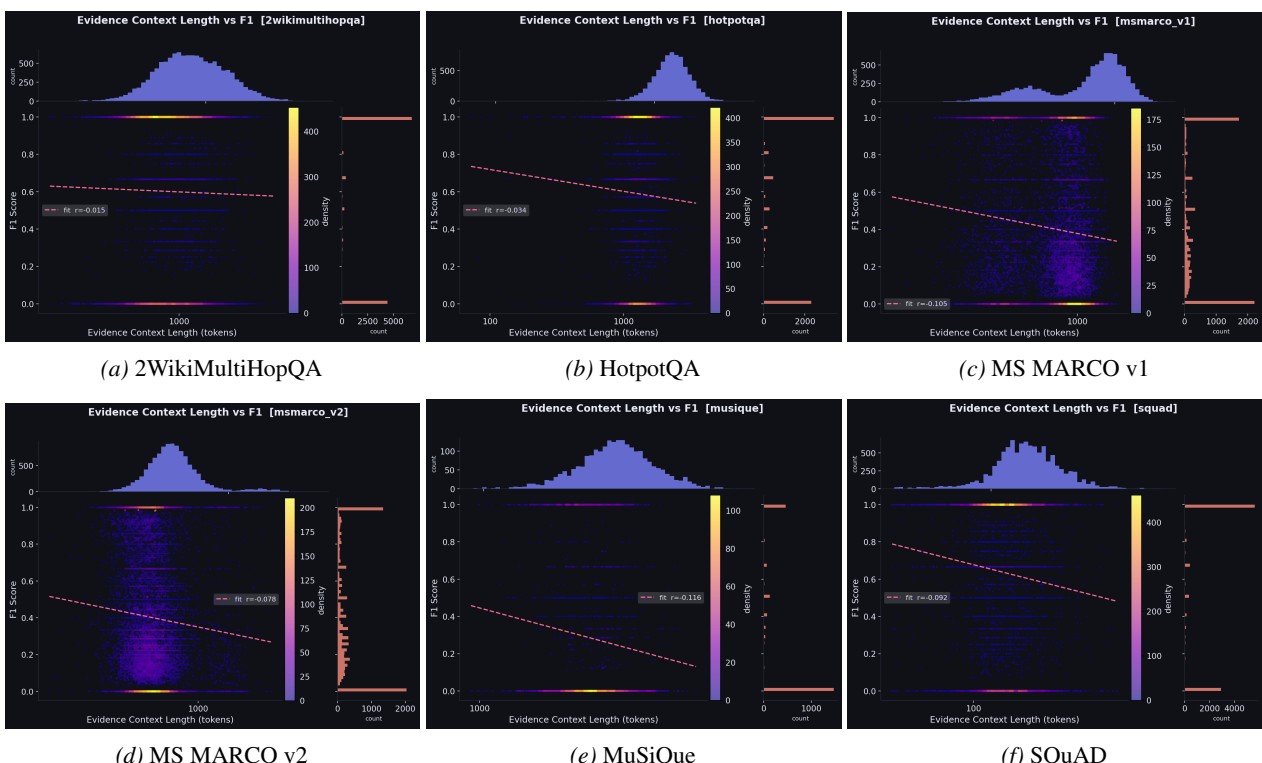

*(a)* 2WikiMultiHopQA        *(b)* HotpotQA        *(c)* MS MARCO v1

*(d)* MS MARCO v2        *(e)* MuSiQue        *(f)* SQuAD

*Figure 16.* Relationship between text length and answer F1-score.

*Table 9.* DoRA results on MS MARCO MQA. Each entry denotes F1-score(Time), where time is measured in seconds.

| num_sample \ num_epoch | 5 | 10 | 20 |
|---|---|---|---|
| 3 | 24.48 (17.0) | 27.90 (34.0) | 36.59 (68.0) |
| 5 | 26.54 (38.4) | 33.69 (56.7) | 48.67 (113.3) |
| 10 | 32.65 (56.7) | 48.94 (113.3) | 56.52 (226.6) |

*Table 10.* IA3 results on MS MARCO MQA. Each entry denotes F1-score(Time), where time is measured in seconds.

| num_sample \ num_epoch | 5 | 10 | 20 |
|---|---|---|---|
| 3 | 24.60 (12.7) | 26.31 (25.38) | 29.06 (50.7) |
| 5 | 25.26 (21.2) | 28.20 (42.3) | 33.58 (84.6) |
| 10 | 27.79 (42.3) | 32.70 (84.6) | 38.25 (169.2) |

## E.5. Multitasks

We fine-tune the hypernetwork after instruction fine-tuning MQA using these four datasets jointly, rather than fine-tuning on each dataset individually. These four datasets represent four different tasks: DailyDialog (Li et al., 2017) maps conversation history to LoRA, GSM8K (Cobbe et al., 2021) evaluates reasoning, Persona-Chat (Zhang et al., 2018) maps preferences and personality to LoRA, and SAMSum (Gliwa et al., 2019) evaluates summarization.

As shown in Table 11, SHINE achieves outstanding results on DailyDialog, Persona-Chat, and SAMSum. The only exception is GSM8K. We believe this is because the instruction fine-tuning data does not contain any reasoning data, and no additional techniques such as reinforcement learning or distillation are used to enhance reasoning ability.

| Method | DailyDialog | GSM8K | Persona-Chat | SAMSum |
|---|---|---|---|---|
| In-Context | 12.28 | 44.48 | 9.32 | 34.01 |
| Naive | 7.85 | 24.70 | 8.97 | – |
| SHINE | 17.12 | 25.27 | 16.07 | 46.18 |

*Table 11.* Performance comparison across four datasets representing different tasks. All using standard F1-Score same as SQuAD evaluation.

## E.6. Scaling

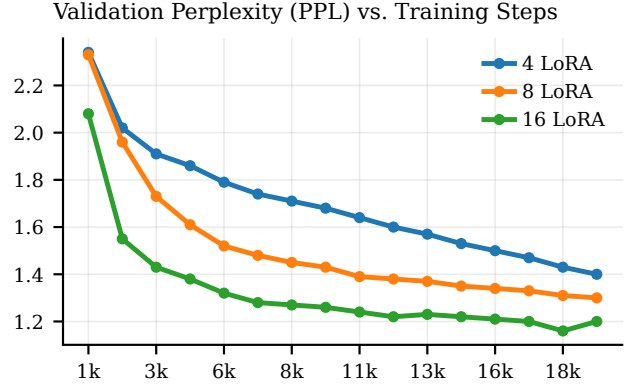

*Figure 17.* **Scaling with Layer Numbers**

*Figure 18.* **Scaling with LoRA Ranks (Memory Token Numbers)**

