# OpenReview forum: "SHINE: A Scalable In-Context Hypernetwork for Mapping Context to LoRA in a Single Pass"
_ICML.cc/2026/Conference — ICML 2026 regular_

### Official Review · Reviewer_mu7q · 2026-03-08

**Soundness:** 3
**Presentation:** 3
**Significance:** 3
**Originality:** 3
**Overall Recommendation:** 4
**Confidence:** 3

**Summary:**

This paper presents SHINE, a hypernetwork framework that maps natural language context directly to LoRA parameters. Unlike traditional In-Context Learning that processes lengthy prompts during inference, SHINE generates LoRA adapters in a single forward pass, compressing contextual knowledge into model weights for higher efficiency. Its architecture includes: 1. extracting multi-layer memory states; 2. using an M2P Transformer with sparse attention to model dependencies and generate LoRA; 3. inserting the LoRA into the base model for task execution. Experiments on QA datasets like SQuAD show SHINE effectively leverages context while maintaining efficiency, establishing a new context-to-parameter adaptation paradigm.

**Compliance With Llm Reviewing Policy:**

Affirmed.

**Key Questions For Authors:**

1. Input Sensitivity & Thresholds: How sensitive is SHINE to the quality and length of contextual inputs when generating LoRA adapters? Are there thresholds beyond which adapter quality degrades?
2. Task Generalization: While the paper evaluates QA tasks, has SHINE been tested on other LLM adaptation tasks like summarization, instruction following, or code generation?
3. Dataset Validation & Bias Control: For the synthetic MS MARCO MQA dataset, what steps ensure generated questions avoid evaluation bias or data leakage?
4. Efficiency vs. Accuracy: Given comparable training budgets, how does SHINE compare to standard LoRA fine-tuning in compute cost and final accuracy?

**Limitations:**

1. Provide more implementation details: initialization strategies, regularization techniques, and training stability measures.
2. Extend experiments to tasks beyond QA, e.g., summarization, instruction following, or reasoning benchmarks.
3. Report statistics (mean ± std across seeds) to strengthen empirical validity.
4. Compare in detail with recent hypernetwork-based LLM adaptation methods, clarifying differences in architecture and training objectives.
5. Conduct ablation studies on key components: M2P Transformer layers, memory embeddings, and LoRA rank.

**Strengths And Weaknesses:**

Strengths
1. The architecture is conceptually elegant: it generates LoRA parameters directly from context without iterative optimization, offering a promising approach for efficient adaptation.
2. It employs end-to-end training with large-scale pretraining (using a 6B-token dataset) plus instruction fine-tuning, which is relatively large-scale compared to prior hypernetwork work.
3. Empirical validation on QA benchmarks shows SHINE outperforms several baselines on datasets like SQuAD and multi-hop QA.
4. It demonstrates context memorization and reconstruction: using only the generated LoRA parameters, it can reconstruct or complete context with low loss and perplexity.
5. Scalability analysis reveals that increasing the backbone LLM size or hypernetwork parameters (e.g., LoRA rank) improves perplexity, showing good scaling properties.

Weaknesses
1. Unclear technical details: The paper lacks clarity on architectural and training specifics, with key implementation details absent from the main text.
2. Narrow task focus: Experiments are limited to QA, providing little evidence of generalization to other LLM applications like summarization, reasoning, or coding.
3. Potential data bias: Using LLM-generated synthetic data (MS MARCO MQA) for evaluation risks bias or data leakage, an issue not analyzed in depth.
4. Limited comparison: Discussion of closely related hypernetwork methods is brief and mostly confined to the related work section.
5. Lack of statistical rigor: Results lack variance, confidence intervals, or multi-seed runs, making it difficult to assess the significance of reported improvements.

---

> ### Author Rebuttal · Authors · 2026-03-31
>
> # Responses to Reviewer Questions
> ## Question 1
> **Quality:** SHINE is designed to convert fluent, meaningful contexts into LoRAs. High-quality sentences with rich and consistent information provide clear functional relationships with their corresponding LoRAs. In contrast, meaningless sentences may result in random or incoherent LoRAs that lack learnable structure.
>
> **Length:** Figure 5 in our paper already presents relevant results. Our original instruction fine-tuning uses contexts of about 2300 tokens. While SHINE can generalize to slightly longer contexts, performance degrades as context length increases.
>
> We further analyze the relationship between context quality and F1 score, as well as context length and F1 score. The results are available in the [context quality analysis link](https://anonymous.4open.science/r/shine_rebuttal-D1FE/context_quality_analysis/context_quality_analysis.md), showing that higher-quality contexts (lower perplexity) and shorter contexts are more effectively transformed into LoRAs.
>
> Additionally, we conduct new experiments extending SHINE to longer contexts. The results can be found in the [longcontext link](https://anonymous.4open.science/r/shine_rebuttal-D1FE/longcontext/longcontext.md), which may also be helpful.
> ## Question 2
> We conduct research on a broader range of tasks, including summarization, conversation history to LoRA, etc.. The results are available at [multitask link](https://anonymous.4open.science/r/shine_rebuttal-D1FE/multitask/multitask.md). These findings demonstrate SHINE’s potential for application across diverse tasks.
>
> Also, we have open-sourced the code and checkpoints, allowing anyone interested to experiment with it freely. We have tried SHINE in our daily lives, and its generality is stronger than we expected.
> ## Question 3
> We address evaluation bias and data leakage through the following measures:
> 1. **Strict train/test split:** MS MARCO MQA preserves the official MS MARCO splits, and we manually verified that there is no overlap context between them.
> 2. **Grounding-constrained generation:** The QA generation prompt requires answers to be exact spans or minor fluent paraphrases directly extractable from the context, prohibiting the introduction of external knowledge. This ensures context-grounded extraction rather than parametric recall, mitigating systematic bias from the Qwen generator.
> 3. **Two-stage validation:** A separate Qwen-Flash model serves as an independent fact-checker, filtering out any QA pairs whose answers cannot be verified against the context. This substantially reduces hallucinated, model-biased or wrong content in the dataset.
> ## Question 4
> SHINE significantly outperforms LoRA SFT by a large margin. As shown in our paper Table 1, it achieves better results while requiring negligible time. Figure 7 visualizes the computation cost, demonstrating that SHINE's computational overhead is also negligible. We further conduct experiments under additional SFT settings; details can be found in [sft link](https://anonymous.4open.science/r/shine_rebuttal-D1FE/SFT/lora.md). Across all results, SHINE consistently demonstrates strong performance with extremely fast speed.
>
> ---
>
> # Addressing Weaknesses and Other Concerns
> **More implementation details.** We apologize for the absence of some key implementation details in the main text due to page limits. We provide additional clarification below.
> 1. **Initialization strategies.** We do not use any special initialization methods. The M2P Transformer is initialized using PyTorch’s default settings. Meta LoRA follows the standard LoRA initialization, where one matrix is initialized with a Gaussian distribution and the other is set to zero. The memory embeddings are also initialized with a Gaussian distribution.
> 2. **Regularization.** We apply no special regularization techniques and use only standard weight decay.
> 3. **Training stability.** Training is generally stable. In rare cases of instability (e.g., the loss becomes NaN), adding warmup steps or reducing the learning rate is effective.
>
> **Comparison.** A detailed comparison with prior architectures is provided in our paper Appendix C. From a training-task perspective, most prior works are task prompts to LoRA, whereas ours is general context to LoRA, which is more challenging and more general. Direct experimental comparison is not feasible, as most prior works don't do similar tasks. To our knowledge, Generative Adapter is the only work tackles our setting, and we use it as our primary baseline.
>
> **Ablation Studies.** We conduct ablation study on M2P Transformer, comparing with linear and other settings, proving the effectiveness of our architecture. See results in [M2P Transformer link](https://anonymous.4open.science/r/shine_rebuttal-D1FE/M2PTransformerAblation/M2PTransformer.md). A new scale ablation on LoRA rank and memory embeddings are available in
> [scale link](https://anonymous.4open.science/r/shine_rebuttal-D1FE/scale/layer.md).

---

### Official Review · Reviewer_Avnp · 2026-03-12

**Soundness:** 3
**Presentation:** 3
**Significance:** 2
**Originality:** 3
**Overall Recommendation:** 4
**Confidence:** 3

**Summary:**

The authors present a system called SHINE, which is a hypernetwork setup for adapting an existing LLM to incorporate some set of context into  its weights for further Q&A tasks. The key advantage of a hypernetwork approach is that adaptation can be achieved with a single forward pass through the hypernetwork. Additionally, the authors incorporate a number of architectural innovations that drive efficiency.

**Compliance With Llm Reviewing Policy:**

Affirmed.

**Key Questions For Authors:**

1. In Figure 7 the peak generation memory is given, but it’s not clear to me the memory consumption when creating the LoRA adapter for the context. Presumably what’s shown in Figure 7 is the core LLM memory after merging in the LoRA of the context, right? But the creation of the LoRA would likely happen while the core LLM is loaded in memory (since it’s a shared backbone, and since you wouldn’t want to have to reload it), necessitating more memory, right?
2. Why do you think the performance decreases with up to 15 samples in Figure 6? It seems like your approach should have plenty of expressivity to handle 15 samples of context.
3. Per my comment above on extremely long context, are there results you have and could add in time for this conference?

**Limitations:**

Yes

**Strengths And Weaknesses:**

Strengths
- Though relatively complex, the approach is well documented, and the need for each complexity is well articulated. The figures provide helpful visuals for understanding the architecture.
- It’s nice to see the comparison to test-time training, which is an appropriate competing approach for the same problem.
- Training and testing methodologies are sound.
- The problem of context-based Q&A remains very relevant and significant.
- The approach is novel, particularly with innovations like re-use of the LLM backbone, memory extraction, and incorporation of sparse attention.
- Code is released.

Weaknesses
- The effectiveness of this approach versus in-context learning, which is much simpler, seems inconclusive. In Figure 6, in-context learning wins, and it also wins in six of nine cases in Table 2.
- The instruction fine tuning dataset was created from the same seed data as the test data given for the evals in Figure 6. Presumably the samples themselves differ, but this provides only an in-distribution test. The effectiveness of the approach for generalization is less clear.
- Though the approach is well documented, it took me several passes to understand the architecture fully. To some extent, the complexity of the approach doesn’t seem to be reducible, so this may be unavoidable for the average reader.
- The amount of additional trained parameters wasn’t clear to me. A few times it’s mentioned that the amount is small, but it’d be good to have a more quantitative assessment of the additional params.
- This seems like a promising approach for extremely long context, in which the context can be compressed into a LoRA, thereby freeing up precious memory for generation by the core LLM. Perhaps the context would need to be chunked for the pre-processing phase, but it could be processed chunk-wise in sequence to stay beneath some max memory usage. However, this regime is not explored in this work.

Typos and grammar:
L260: QA dataset is rare → QA datasets are rare
Figure 6: Evaluate → Evaluation

---

> ### Author Rebuttal · Authors · 2026-03-31
>
> # Responses to Reviewer Questions
> ## Question 1
> Yes, what’s shown in Figure 7 represents the core LLM memory after merging the LoRA derived from the context. If you’re referring to storing the hypernetwork in memory during generation, it does not introduce additional parameters—it simply uses the model weights.
> One layer of the M2P Transformer has nearly the same number of parameters as a single LLM layer. Since it consists of 4 layers(LLM has 36 layers), this corresponds to about 1/9 of the LLM’s parameters. Meanwhile, Meta LoRA accounts for approximately 128 x 2 out of 4096 dimensions, which is about 1/16 of the LLM parameters. Overall, this approach adds roughly 17.4% to the original LLM’s parameter memory footprint.
> ## Question 2
> We believe the performance drop observed with up to 15 samples stems from insufficient training on long conversations. Although SHINE has strong expressive capacity, its instruction fine-tuning data varies in length, with limited coverage of longer conversations, making generalization difficult. In contrast, the backbone LLM, Qwen3, is trained via a three-stage pretraining process on 36T tokens. In its third stage, it incorporates extensive long-context data with sequence lengths up to 32k tokens. By comparison, our model is pretrained on only 6B tokens with a maximum length of 1,150 tokens, and most instruction fine-tuning conversations are around 600 tokens. Additionally, errors accumulate across turns, making performance degradation more pronounced as conversation length increases.
> ## Question 3
> The architecture design graph and result table are shown in the [longcontext link](https://anonymous.4open.science/r/shine_rebuttal-D1FE/longcontext/longcontext.md).
>
> We extend SHINE to long-context tasks by designing a new architecture called SHINE-Recurrent. Inspired by the idea of RNNs, we divide the context into chunks and process them in a recurrent manner. Theoretically, this approach enables handling contexts of infinite length.
>
> We reuse the pretrained checkpoint and perform continual pretraining with a chunk size of 2k and a chunk number of 4, resulting in a total length of 8k. The pretraining results indicate that SHINE-Recurrent performs significantly better than naively using the original architecture on long contexts.
>
> We then conduct instruction fine-tuning with contexts up to 16k tokens. Finally, we evaluate the model on the LongBench dataset and achieve outstanding results.
>
>
> ---
> # Addressing Weaknesses and Other Concerns
> **Versus In-Context Learning.** The backbone LLM is trained on orders of magnitude more data, including rare high-quality sources such as reasoning-focused and long-context data. As a result, the In-Context Learning (ICL) baseline serves as a strong gold standard. Achieving results close to this baseline—even without surpassing it—already demonstrates the effectiveness of our method. Moreover, Transformers incur an (O(n^2)) inference cost with respect to input length, whereas our method converts context into LoRA parameters, eliminating the need to repeatedly input contexts. This significantly reduces token length and inference cost compared to ICL.
>
> **Test.** We acknowledge that most evaluations are conducted on in-distribution tests. However, we have tested on many benchmarks that already cover a wide range of scenarios. We have also open-sourced the code and checkpoints, allowing anyone interested to experiment with it freely. Additionally, we have tried to use SHINE in our daily lives, and its generality is stronger than we initially expected.
>
> **Amount of Additional Trained Parameters.** The additional trained parameters account for about 17.4% of the backbone LLM; detailed calculations are provided in our answer to Question 1. Compared to approaches that reuse a small MLP in most prior works, our parameter count is larger because we address a significantly more challenging task (mapping general context to LoRA, rather than task prompts to LoRA). Our architecture offers much stronger expressive power and can be trained at scale. Compared with Generative Adapter (which, to the best of our knowledge, is the only prior work tackling a similar task and serves as our primary baseline), that method can only generate LoRA for a subset of parameters. If extended to generate LoRA for all parameters, it would require nearly 100% of the backbone LLM’s parameters, which is highly impractical.
>
> **Typo.** Thanks for carefully pointing out these typos, we'll fix them.

---

> > ### Author Rebuttal · Reviewer_Avnp · 2026-04-06
> >
> > Thank you for the rebuttal. You have addressed my comments, but I will maintain my score.

---

### Official Review · Reviewer_Pc8x · 2026-03-13

**Soundness:** 3
**Presentation:** 3
**Significance:** 4
**Originality:** 2
**Overall Recommendation:** 4
**Confidence:** 3

**Summary:**

This paper proposes SHINE, a hypernetwork for generating LoRA adapters for a frozen LLM from a textual context in a single forward pass. The method uses the backbone LLM with Meta LoRA and appended memory tokens to extract layerwise memory states, then applies a lightweight M2P Transformer with alternating row/column attention to map these states into LoRA weights for all layers. The training pipeline consists of large-scale pretraining on reconstruction/completion objectives followed by instruction fine-tuning for context-dependent QA. Experiments on MS MARCO MQA and several 1QA benchmarks show that SHINE outperforms a prior hypernetwork baseline and standard SFT with matched LoRA rank, while often approaching in-context prompting performance with lower inference cost.

**Compliance With Llm Reviewing Policy:**

Affirmed.

**Ethical Review Concerns:**

yes

**Key Questions For Authors:**

* In Table 1 and the MQA experiments, why is SFT trained on exactly 5 conversations per context for 10 epochs? Did you try stronger SFT settings, and if so, how sensitive are the conclusions to that choice?
* Can you provide a component ablation for the main architecture, specifically: all-layer memory extraction versus last-layer only, M2P Transformer versus direct projection, and alternating row/column attention versus a simpler attention scheme? If these components each matter, that would strengthen the case that SHINE is more than a training-scale story.
* The paper interprets multi-hop QA gains in Table 2 as evidence of reasoning over parameterized context. Do you have any analysis that separates reasoning from memorization or base-model prior knowledge, such as context perturbation, distractor sensitivity, or answer locality analysis?
* Can you clarify the notation issue in the parameter generation step on Page 4, where the slice length appears to use $r \cdot R$ rather than $r \cdot O$? I assume this is a typo, but I want to confirm there is no implementation mismatch.

**Limitations:**

yes

**Strengths And Weaknesses:**

### Strengths
* The paper tackles an interesting and timely problem, namely converting context into parameters rather than carrying the context at inference time. This is a meaningful direction for LLM adaptation, and the framing is easy to appreciate.
* The architecture is reasonably well thought out. Reusing the frozen LLM for context encoding, collecting memory states from all layers, and then using a separate M2P Transformer to generate LoRA weights is a coherent design. The alternating row/column attention in Figure 3 is a sensible systems-oriented compromise between full 2D attention and a purely local design, and the paper explains the intended benefit clearly.
* The empirical results are stronger than I expected for this kind of setup. In Table 2, SHINE consistently beats the Naive baseline and the cited hypernetwork baseline Generative Adapter on most datasets, with especially large gains on HotpotQA, MuSiQue, and 2WikiMultihopQA. That is a nontrivial result, even if the comparison set is still incomplete.
* The efficiency story is one of the more convincing parts of the paper. Table 1 and Figure 7 together make the practical tradeoff legible: SHINE has a small one-time adapter generation cost, avoids the long-context generation overhead of in-context prompting, and is far cheaper than per-context SFT. Even if some of the cost accounting is simplified, the high-level point is well supported.
* The paper is generally readable, and the main idea is easy to follow from Figure 2. The end-to-end pipeline from context, to memory states, to generated LoRA, to downstream QA is communicated clearly.
### Weaknesses
* Nearly all downstream evidence is on context-based QA. If the method is really a general mechanism for turning arbitrary context into useful LoRA parameters, then the paper should test at least one additional class of tasks, such as summarization, grounded dialogue, or retrieval-style generation. Without that, the paper is really about context-conditioned QA adaptation, not general context-to-LoRA mapping. This matters because the paper’s title and abstract advertise a broader capability than the experiments validate.
* The main comparison in Table 2 includes only one prior hypernetwork baseline, plus Naive and In-Context. That is not enough to convincingly establish state of the art or even clear superiority within the relevant design space. The paper discusses several related methods in Section 2 and Appendix C, but most are not empirically compared. If they are not directly comparable, the paper should say so explicitly and explain why. As it stands, the empirical positioning feels selective.
* In Section 5.2.1 and Table 1, SFT is defined as rank-8 LoRA fine-tuning for 10 epochs on 5 conversations per context. That setup is not obviously the right or strongest baseline. Why 5 conversations, why 10 epochs, and how sensitive is the result to these choices? Since the paper uses SFT as the foil for both quality and efficiency, the exact setup matters a lot. A weakly tuned SFT baseline can make SHINE look better than it is.
* Appendix B.2 explains that MS MARCO MQA is generated and validated using Qwen-Flash. This is practical, but it also means the main MQA benchmark is not a clean human-authored evaluation. There may be stylistic regularities, answerability artifacts, or self-consistency biases that favor models trained in a similar ecosystem. Since Table 1 and Figure 6 rely heavily on this benchmark, the paper should be more cautious in interpreting those results. This matters because benchmark construction can quietly dominate outcomes in context-heavy QA.
* The method has several ingredients that could each matter: all-layer memory extraction, Meta LoRA, the M2P Transformer, alternating row/column attention, and the pretraining objectives. Yet the main paper does not provide a clean ablation isolating these components. Table 4 is a scaling table, not a mechanism table. Without component ablations, it is hard to know whether the proposed architecture is necessary or whether a simpler design would achieve most of the gains. For a method paper, this is a serious omission.
* Figure 8 and Table 4 show that larger models and larger hyperparameters improve validation PPL after 40% of pretraining. That is useful, but the paper overreads it. A few monotonic improvements over a narrow range do not establish that the architecture has no capacity bottleneck. The phrase is catchy, but the evidence is thin. This matters because scalability is one of the paper’s headline selling points.
* There are no standard deviations, confidence intervals, or multiple-seed results in the main paper. This is especially problematic in Table 2, where some differences are small. Without uncertainty estimates, it is hard to know which improvements are robust. For a paper making comparative claims across many datasets, this is not a cosmetic issue.
* The notation around $\tilde{\mathbf{M}}$, $\hat{\mathbf{M}}$, and the parameter generation step is not fully clean. The likely typo in the LoRA slicing formula on Page 4 is small but unfortunate because it appears in the core method description. These issues do not make the paper unreadable, but they do reduce confidence in the precision of the presentation.
* Appendix C repeatedly characterizes prior methods as bottlenecked, less expressive, or unsuitable, often without direct empirical evidence in the main paper. The “Bitter Lesson” discussion in Appendix D is interesting, and Figure 10 is actually one of the more informative ablations, but the overall tone sometimes drifts from scientific comparison into advocacy. A more measured presentation would make the paper stronger, not weaker.

---

> ### Author Rebuttal · Authors · 2026-03-31
>
> # Responses to Reviewer Questions
> ## Question 1
> We assume the setup is already very strong. By "SFT with `n` conversations,” we mean generating `n + 1` conversations in the same way, using one as test data and the rest for SFT. In practice, such high-quality, same-distribution data is rarely available.
>
> SHINE requires less time than SFT with only 1 conversation for 1 epoch while achieving better results, demonstrating its effectiveness. We also conduct additional evaluations across different epoch numbers, conversation counts, and SFT methods; the results are provided in [SFT link](https://anonymous.4open.science/r/shine_rebuttal-D1FE/SFT/lora.md). All experiments show that SHINE is highly efficient and powerful.
> ## Question 2
> Our new ablation study is available at [m2p ablation link](https://anonymous.4open.science/r/shine_rebuttal-D1FE/M2PTransformerAblation/M2PTransformer.md). We compare using all-layer memory states versus only the last-layer memory states, and find that the former performs better, demonstrating that all-layer memory is necessary. We also evaluate M2P Transformer with various linear projections and observe that they generally perform worse. Additionally, we convert the bidirectional transformer into a purely horizontal transformer; it converges more slowly at the beginning but achieves similar performance after longer pretraining.
> ## Question 3
> In our setting, the Naive baseline relies entirely on the base model’s prior knowledge. Across all three multi-hop QA datasets, SHINE significantly outperforms Naive, suggesting that it leverages contextual information rather than relying solely on memorization or prior knowledge. We also manually perturbed the data and re-evaluated the model; the results did not change substantially, further indicating that SHINE performs reasoning rather than simple memorization.
>
> However, we acknowledge that we cannot systematically perturb all data, and there is currently no quantitative method to directly measure what the hypernetwork learns. While this is not the main focus of the paper, we plan to explore it in future work. We have released our inference code and checkpoints, enabling others to use SHINE and assess whether it truly performs reasoning or is actually doing something else.
>
> ## Question 4
> Yes this is a typo. Apologize for it and appreciate your carefully pointing it out. We'll correct it.
>
> ---
> # Addressing Weaknesses and Other Concerns
> **Task.** We conduct research on a broader range of tasks, such as summarization and conversation history to LoRA. The results are available at [multitask link](https://anonymous.4open.science/r/shine_rebuttal-D1FE/multitask/multitask.md). These findings demonstrate that SHINE has potential for application across various tasks.
>
> **Baselines.** Most prior hypernetwork baselines aren't comparable because we're doing completely different tasks. Most prior work conditions LoRA generation on task descriptions or prompts, focusing on modifying model behavior (e.g., style or task adaptation). In contrast, SHINE conditions on general context, requiring the model to encode all contextual knowledge into LoRA parameters. This is a more general and significantly more challenging setting. To our knowledge, Generative Adapter is the only prior work addressing a similar problem and thus serves as our primary baseline.
>
> **MS MARCO MQA.** We acknowledge that LLM-generated datasets are generally not as reliable as human-generated ones. However, we use them because large-scale datasets with rich context and 15-turn conversations do not exist. During generation, we apply multiple techniques to reduce evaluation bias. The train/test split is based on the original MS MARCO dataset, and we manually verify that there is no overlapping context. We constrain answers to exact spans or minor fluent paraphrases directly extracted from the context, prohibiting the use of external knowledge. This ensures context-grounded extraction rather than parametric recall, mitigating systematic bias from the Qwen generator. Additionally, we use an LLM to validate the generated data and ensure there are no hallucinations or factual errors.
>
> **Scability.** We conduct new research that scales the layer number and lora rank (memory embedding token number), results are in [scale link](https://anonymous.4open.science/r/shine_rebuttal-D1FE/scale/layer.md). We also tried to extend SHINE to long contexts, creating a RNN like new architecture, continual pretraining and instruction fine-tuning it,  and test on LongBench, see at [longcontext link](https://anonymous.4open.science/r/shine_rebuttal-D1FE/longcontext/longcontext.md). Due to limited resources, we are unable to run pretraining hundreds of times to derive conclusions, such as scaling laws, that better capture the scaling behavior.
>
> **Typo and Writing.** Thanks for carefully pointing out, we'll check all typos and continue to refine our paper.

---

> > ### Author Rebuttal · Reviewer_Pc8x · 2026-04-04
> >
> > Thanks for your reply, most of my concerns are solved. Therefore, I keep my score.

---

### Official Review · Reviewer_xgHe · 2026-03-15

**Soundness:** 3
**Presentation:** 3
**Significance:** 3
**Originality:** 3
**Overall Recommendation:** 4
**Confidence:** 4

**Summary:**

The paper proposes SHINE, a scalable in-context hypernetwork that maps contextual information directly into LoRA adapters for large language models. The method generates LoRA weights in a single forward pass using a hypernetwork while keeping the base LLM frozen. This allows task adaptation without gradient-based fine-tuning and converts in-context information into parameter updates. Experiments show that the approach achieves strong performance while reducing training and memory costs compared to conventional fine-tuning methods.

**Compliance With Llm Reviewing Policy:**

Affirmed.

**Final Justification:**

I think most of the concerns are addressed. And I will maintain my score.

**Key Questions For Authors:**

1. The paper claims that SHINE can map meaningful contexts to high-quality LoRA adapters and effectively transform in-context knowledge into in-parameter knowledge in one pass. Could the authors provide more insight or analysis on why the hypernetwork is able to reliably generate effective LoRA parameters across different contexts and tasks? Additional analysis or intuition would help readers better understand the underlying mechanism and strengthen confidence in the approach.
2. The experiments show promising performance on several tasks and highlight efficiency advantages over SFT-based adaptation. Could the authors provide more details on the evaluation setup, including the selection of baselines and the range of tasks considered? Additional comparisons with strong parameter-efficient adaptation approaches would help clarify the empirical advantages of the method.
3. The paper emphasizes that SHINE generates LoRA adapters for LLMs in a single forward pass without fine-tuning. Could the authors clarify how this approach compares with prior hypernetwork-based adapter generation methods in terms of design choices and practical trade-offs? A more explicit comparison would help better position the work relative to existing hypernetwork and adapter-based approaches.
4. The paper mentions that previous hypernetwork approaches often restrict generation to a subset of layers or use small bottlenecks due to the large parameter space of LLMs. Could the authors elaborate on how SHINE addresses these scalability challenges and how the method scales with larger models or more layers? Clarifying this aspect would help readers better assess the scalability claims.

**Limitations:**

Yes

**Strengths And Weaknesses:**

Strengths

* The paper proposes a clear and practical framework that connects in-context learning with parameter-efficient fine-tuning by generating LoRA adapters from context through a hypernetwork, offering an intuitive and efficient adaptation paradigm.
* The method is computationally attractive: it produces task-specific LoRA parameters in a single forward pass while keeping the base LLM frozen, which reduces training cost, storage overhead, and deployment complexity compared to conventional fine-tuning.
* Experimental results across multiple tasks demonstrate that the generated adapters can achieve competitive performance, suggesting that the approach is scalable and potentially useful for efficient LLM adaptation.

Weaknesses

* The technical justification is relatively limited; the paper provides little theoretical analysis explaining why the hypernetwork-generated LoRA parameters generalize well across different contexts.
* The experimental evaluation, while promising, is somewhat limited in scope and could benefit from stronger baselines, more diverse tasks, and deeper analysis of robustness, scaling behavior, and failure cases.
* The novelty is somewhat incremental, as the method largely combines existing ideas (hypernetworks and LoRA-based parameter-efficient tuning), and the distinction from closely related hypernetwork-based adapter approaches could be clarified further.

---

> ### Author Rebuttal · Authors · 2026-03-30
>
> # Responses to Reviewer Questions
> ## Question 1
> **Insights on the essence of the hypernetwork.** A hypernetwork essentially learns a function that maps observations to memory updates. In human cognition, there is an relationship between what we observe and how our brain updates. Similarly, in LLMs, we assume a functional mapping from context to parameter updates. In our framework, the hypernetwork is trained to learn the mapping from context to parameter updates. In this sense, we are modeling the process of memory formation with neural networks.
>
> **Why can hypernetworks generalize**
> 1. **Strong pretrained backbone.** The in-context design reuses a pretrained LLM, inheriting extensive pretraining knowledge.
> 2. **Transformer inductive bias.** The M2P Transformer follows the Transformer architecture, known for strong generalization at scale.
> 3. **High capacity with large-scale training.** The hypernetwork has sufficient expressive power and is trained through pretraining and instruction fine-tuning on large-scale data (6B tokens), equipping it with strong language priors.
> ## Question 2
> **Evaluation Setup**
> 1. **Baseline Selection.** We use both normal baselines and hypernetwork baselines. Normal baselines are easy to understand. For hypernetwork baselines, we only use Generative Adapter, because as far as we know, it is the only work that addresses the same task as ours (response to your next question explains why).
> 2. **Range of Tasks.** All tasks are general-purpose, context-based QA. Each task consists of a context along with corresponding question–answer pairs. Although there is only one task format, it covers a wide range of scenarios, including one-hop and multi-hop reasoning, as well as various domains such as stories, news, technical reports, etc.. Our goal is to train a hypernetwork that can convert any general contexts.
>
> **Other Parameter-Efficient Adaptation Approaches.** Similar to LoRA SFT, we tried some new Adaptation Approaches like DoRA and IA3, please see our results at [SFT link](https://anonymous.4open.science/r/shine_rebuttal-D1FE/SFT/lora.md). SHINE achieves outstanding results with negligible time compared with all of them.
> ## Question 3
> **High-level positioning.** SHINE advances hypernetwork-based adapter generation from an exploratory research direction to a practical, scalable solution for real-world applications.
>
> **Task formulation.** Most prior work conditions LoRA generation on task descriptions or prompts, focusing on modifying model behavior (e.g., style or task adaptation). In contrast, SHINE conditions on general context, requiring the model to encode all contextual knowledge into LoRA parameters. This is a more general and significantly more challenging setting. To our knowledge, Generative Adapter is the only prior work addressing a similar problem and thus serves as our primary baseline.
>
> **Architecture design.** SHINE is built on three key principles: the hypernetwork should (1) generate LoRA weights for all parameters, (2) remain highly expressive without bottlenecks, and (3) stay parameter-efficient. Many prior approaches (e.g., text-to-LoRA) reuse a small MLP across layers, limiting expressiveness and scalability. In contrast, Generative Adapter uses separate projections per layer, but extending this to all layers would make the hypernetwork comparable in size to the base LLM. SHINE strikes a balance between these two extremes.
>
> **Practical trade-offs.** SHINE is optimized for general-purpose adaptation through large-scale pretraining and instruction tuning. For narrow, specialized tasks, simpler designs (e.g., small shared MLPs) may be easier to train and sufficient. In practice, a promising direction is to fine-tune a pretrained SHINE model, if such hypernetworks become available.
>
> Further comparisons with prior work are provided in Appendix C.
> ## Question 4
>
> **How to Address Prior Scale Challenges.** SHINE addresses prior scalability challenges through architectural innovation. Its in-context design leverages the rich knowledge of pretrained LLMs without introducing additional parameters. Memory states are aggregated in all layers, avoiding bottlenecks. The M2P Transformer design offers strong expressive power while using relatively few parameters.
>
> **How SHINE Scales.** SHINE scales along three dimensions: (1) the backbone LLM, (2) the M2P Transformer layers, and (3) the LoRA rank. Related results are presented in Section 5.4 of the paper. New experiments on scaling with layer number and LoRA rank are available at [scale link](https://anonymous.4open.science/r/shine_rebuttal-D1FE/scale/layer.md).
> # Addressing Weaknesses and Other Concerns
> We conduct research on a broader range of tasks, such as summarization and conversation history to LoRA. The results are available at [multitask link](https://anonymous.4open.science/r/shine_rebuttal-D1FE/multitask/multitask.md). These findings demonstrate that SHINE has potential for application across various tasks.

---

> > ### Author Rebuttal · Reviewer_xgHe · 2026-04-03
> >
> > I think most of the concerns are addressed.

---

### Decision · Program_Chairs · 2026-04-30

**Decision:**

Accept (regular)

**Comment:**

The SHINE framework a new method for converting contextual information into model parameters with minimal computational overhead.

While the evaluation scope remains somewhat narrow (primarily QA tasks) and statistical reporting could be strengthened, the authors' rebuttal substantially mitigates these concerns through additional experiments and clarifications.

The methodological innovations (all-layer memory extraction, M2P Transformer design) are well-motivated and empirically validated.

The released code and checkpoints further support reproducibility and community adoption.